# PhenoKG: Knowledge Graph-Driven Gene Prioritization and Patient Insights from Phenotypes Alone

## Abstract

Rare genetic disorders affect over 300 million people worldwide and remain difficult to diagnose, with current genomic approaches yielding definitive answers in only 30-50% of cases. Existing phenotype-driven methods often rely on expert-curated candidate gene lists and are sensitive to incomplete clinical data, limiting their real-world utility. We present PhenoKG, a knowledge-graph framework that enriches patient phenotypes with biomedical knowledge and can rank flexible amount of genes by their likelihood of being causative($\sim$4,000). PhenoKG integrates graph neural networks with transformer-based encoders to capture patient-specific phenotype-gene relationships, and incorporates an optional reranking procedure that leverages recently validated clinical associations to extend the knowledge graph while maintaining robustness to noisy or incomplete input. Designed to operate with or without candidate lists, PhenoKG achieves strong performance across diverse diagnostic settings and consistently outperforms state-of-the-art methods on rare disease benchmarks. Together, these results position PhenoKG as a step toward scalable, phenotype-first models for rare disease diagnosis, and open the path to integrating heterogeneous biomedical data for faster, more equitable genetic discovery.

## 1 Introduction

Rare genetic disorders affect more than 300 million people worldwide and pose major challenges for healthcare due to their low prevalence, heterogeneous symptoms, and limited diagnostic expertise (Nguengang Wakap et al., 2020). Although a typical genome contains tens of thousands of differences from the reference sequence, clinical pipelines usually narrow the search space to a limited set of candidate genes that may explain the phenotype. Distinguishing the causative gene from this large background requires linking patient phenotypes to genes with established or plausible disease associations. These delays hinder timely treatment and increase the burden on patients and families, and especially harmful in children, where most rare diseases appear early and missed diagnoses can lead to irreversible complications.

Seminal studies have shown that identifying the gene that causes the disorder improves diagnostic accuracy and informs prognosis, counseling, and therapeutic decisions. Clinicians integrate heterogeneous evidence—phenotypic abnormalities, their evolution, laboratory findings, and sequencing data—to determine the underlying genetic cause of a patient's condition (James et al., 2016). While sequencing has become relatively inexpensive and widely available through national programs and hospital pipelines, interpretation remains the critical bottleneck. A typical genome contains tens of thousands of differences from the reference genome called variants, most of which are harmless. Distinguishing the few that are potentially disease-causing requires linking variants to the set of observable traits and symptoms named as patient's phenotype. This process that is still largely manual and depends on expert-curated candidate gene lists, considered plausible given current medical knowledge. These lists vary across institutions and risk missing causative genes that are poorly annotated, recently discovered, or filtered out during variant prioritisation.

From a machine learning perspective, this constitutes a highly imbalanced and noisy inference task: the signal is sparse, phenotype descriptions are incomplete or ambiguous, and the hypothesis space

is vast (on the order of $\sim$20,000 genes). Existing phenotype-driven approaches provide valuable support but often degrade under realistic clinical conditions and fail to exploit the broader relational structure connecting phenotypes, genes, and diseases.

When a gene with a well-established disease association matches the patient's phenotype spectrum, variants in that gene can often be classified as pathogenic under ACMG/AMP and ClinGen guidelines (Richards et al., 2015; Strande et al., 2017). However, such assessments depend on expert curation and up-to-date annotations, which are often lacking. This motivates complementary approaches that operate at the gene level to guide clinicians toward plausible candidates even before variant interpretation.

This gap underscores the need for next-generation machine learning frameworks—particularly graph-based and multimodal approaches that can prioritise causative genes directly from phenotype data, independent of candidate lists, while remaining robust under realistic clinical conditions (Decherchi et al., 2021).

We introduce **PhenoKG**, a scalable knowledge-graph framework for phenotype-driven gene prioritisation. Starting from patient phenotypes, PhenoKG integrates structured biomedical knowledge and learns patient-specific graph representations to rank approximately 4,000 genes by their likelihood of being causative. Unlike previous methods, PhenoKG is designed to operate both with and without candidate gene lists, making it robust across diverse clinical workflows. It's design further supports integration of emerging biomedical resources and improves resilience to noisy input.

Our contributions are as follows:

1. We propose PhenoKG, a novel patient-centric knowledge-graph model that captures phenotype-gene relationships and outperforms state-of-the-art methods on multiple rare disease benchmarks.

2. We demonstrate strong performance in both candidate and non-candidate settings, reflecting realistic diagnostic conditions.

3. We introduce a reranking strategy that leverages validated phenotype-gene associations, improving robustness to incomplete data while extending the coverage of existing knowledge graphs.

## 2 RELATED WORKS

To address the diagnostic odyssey in rare genetic disorders, prior work has followed three main machine learning strategies: *genotype-based*, *phenotype-based*, and *hybrid* approaches.

**Genotype-based methods.** These approaches leverage genomic sequencing data to identify variants associated with disease. Common techniques include variant frequency analysis, pathogenicity prediction, and use of curated gene–disease databases. Representative tools include MutationTaster (Steinhaus et al., 2021), CADD (Rentzsch et al., 2019), and M-CAP (Jagadeesh et al., 2016), which predict variant pathogenicity. While effective when comprehensive variant annotations are available, these methods are limited by incomplete training data and provide little guidance in the absence of sequencing.

**Phenotype-based methods.** Phenotype-driven tools prioritize genes or diseases by comparing a patient's phenotypic abnormalities to curated knowledge bases. Early examples include Phenomizer (Ullah et al., 2013), Phenolyzer (Yang et al., 2015), and Phrank (Jagadeesh et al., 2019). More recent efforts exploit facial phenotyping, such as DeepGestalt (Gurovich et al., 2019), GestaltMatcher (Hsieh et al., 2022), and PEDIA (Hsieh et al., 2019), which map visual features to candidate genes. In parallel PhenoApt (Chen et al., 2022) integrates HPO (Köhler et al., 2019), OMIM (Hamosh et al., 2005), and Orphanet (Nguengang Wakap et al., 2020) into a heterogeneous knowledge graph and learns embeddings to rank genes by similarity to a patient's phenotype profile. Complementary to these approaches, Amelie (Birgmeier et al., 2020) uses natural language processing to mine biomedical literature and score genes in a candidate list. One of the mostly used by clinicians method is rule-based Phen2Gene method (Zhao et al., 2020), that ranks disease-linked genes by scoring matches between patient HPO terms and precomputed phenotype–gene associations, providing fast

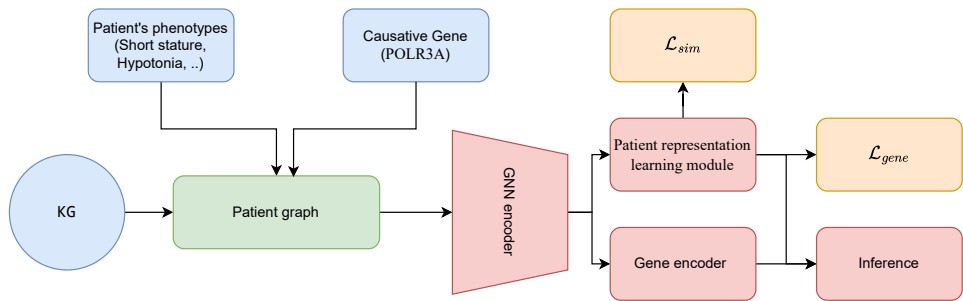

Figure 1: Overview of PhenoKG for rare disease gene prioritization. The model constructs a patient-specific subgraph using phenotypes and a knowledge graph. GATv2 layers generate node embeddings, which are used to create patient and gene representations. These feed into loss functions to predict the most likely causative gene.

candidate prioritization without requiring sequencing. Despite these advances, such approaches have advanced phenotype-driven prioritisation but remain sensitive to incomplete or noisy input.

**Hybrid methods.** Hybrid strategies combine genotype and phenotype evidence, often through probabilistic integration (Robinson et al., 2020; Javed et al., 2014) or deep learning (Smedley et al., 2015; Li et al., 2019; Yoo et al., 2021; Anderson et al., 2019). A recent example, AI-MARRVEL (AIM) (Mao et al., 2024), uses random forests to jointly rank variants from sequencing data and phenotype terms. These approaches improve diagnostic accuracy but still require high-quality sequencing data as input.

**Knowledge-graph methods.** Several works explicitly model phenotype–gene–disease associations as graphs. CADA (Peng et al., 2021) builds a phenotype–gene knowledge graph(KG) and applies Node2Vec embeddings to rank candidate genes, but underperforms compared to more recent methods. Shepherd (Alsentzer et al., 2025) introduces shared pre-training across disease prediction, gene prioritisation, and "patient-like-me" tasks. For gene prioritisation, it trains a KG encoder via link prediction followed by a ranking model. A common limitation of most KG-based approaches, including CADA and Shepherd, is that they rely on fixed, pre-constructed graphs and do not provide mechanisms to dynamically incorporate new phenotype–gene or disease–gene associations during inference. In addition, Shepherd specifically requires the presence of a candidate gene set, curated by experts or generated through external tools. This restricts its applicability in settings without sequencing data or expert knowledge.

Unlike genotype-based and hybrid methods, our approach does not require sequencing data. Compared to existing phenotype-based and KG-based methods, PhenoKG operates directly from phenotype terms, enriches them with biomedical knowledge, and ranks genes without requiring an external candidate list. This makes it broadly applicable across diverse diagnostic scenarios.

## 3 METHOD

We propose PhenoKG, a framework for predicting the causative gene in rare monogenic diseases by prioritizing genes associated with patient phenotypes. Our approach leverages the PrimeKG knowledge graph $G$ (Chandak et al., 2023) to model gene-phenotype relationships. Specifically, PhenoKG learns informative representations from patient-specific subgraphs $G_p$, which uniquely represent a patient based on their list of phenotypes and a set of candidate genes provided by clinicians or obtained from the knowledge graph. The patient subgraph is further augmented with additional information from the knowledge graph, such as related diseases and other relevant entities. As output, the model provides a ranked list of candidate genes, each associated with a relevance score indicating its likelihood of being causative for the patient. An overview of the proposed method is shown in Figure 1.

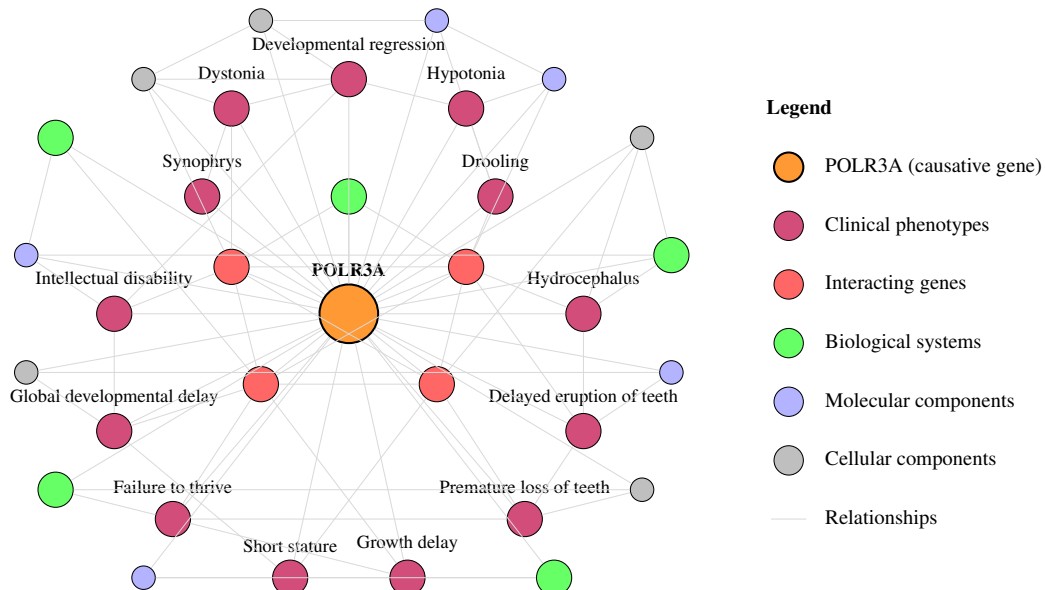

Figure 2: The example of a simplified patient-specific one-hop subgraph and its neighborhood, illustrating the relationships between the *POLR3A* gene and associated genes, diseases, and phenotypes. The real patient subgraph maintains a similar structure but is substantially more complex.

PhenoKG builds on Shepherd's insight of leveraging a biomedical knowledge graph, but introduces key framework-level methodological innovations: (1) patient-specific phenotype-gene subgraph construction, enabling inference with candidate gene lists and large gene search spaces; (2) a Transformer-based gene encoder that jointly processes gene embeddings within each patient graph; and (3) inference-time dynamic reranking, allowing integration of new phenotype–gene evidence without retraining. In contrast, Shepherd relies on embeddings pretrained on the global KG and requires a fixed candidate gene set at inference time, without extracting patient-tailored subgraphs from the KG. These distinctions define PhenoKG as a new and more flexible framework for phenotype-driven gene prioritization.

### 3.1 PROBLEM FORMULATION

Given a patient $p$ with an associated set of phenotypes $\mathcal{P}_p$, encoded using the Human Phenotype Ontology (HPO) system (e.g., [HP:0004322, HP:0001263, ...]), and a complete candidate gene set $\mathcal{G}$ (e.g., [ENSG00000123066, ENSG00000141510, ...]), the set $\mathcal{G}$ with the goal of identifying the causative gene $g^* \in \mathcal{G}$ responsible for the patient's condition (e.g., $g^* = $ ENSG00000165588, which corresponds to the *OTX2* gene).

For each patient, the candidate gene list can be extracted from the global knowledge graph $G$, either based on phenotype-gene associations or as specified by clinicians. The global knowledge graph $G$ is represented as an undirected graph defined by $G = (X, E, A)$, where $X \in \mathbb{R}^{N \times d_{\text{in}}}$ denotes the set of node features, with $N$ being the number of nodes in KG and $d_{\text{in}}$ the feature dimension. Each node, corresponding to a gene, phenotype, or other biomedical entity, is initialized with a unique embedding obtained by pretraining $G$ on a link prediction task, as described in (Alsentzer et al., 2025). The set $E$ represents the edges capturing relationships between nodes, such as phenotype-gene associations and gene-gene interactions. The matrix $A$ defines the connectivity structure of the graph and is described in detail in Section 4.1.

### 3.2 PATIENT-SPECIFIC SUBGRAPHS

To construct the patient-specific subgraph $G_p$, we compute the shortest paths from each phenotype $p_i \in \mathcal{P}_p$ to a provided or inferred list of candidate genes. The subgraph $G_p$ includes all nodes along these paths, along with any additional nodes required to ensure connectivity. If no candidate gene

list is available, the $k$-hop neighborhood can be extracted for each phenotype from $G$, and all genes within this neighborhood are treated as candidates. Formally, $G_p$ is represented as $(X_p, E_p, A_p)$, where $X_p \subseteq X$, $E_p \subseteq E$, and $A_p \subseteq A$. Here, $X_p \in \mathbb{R}^{n \times d_{\text{in}}}$ denotes the node feature matrix, $E_p \subseteq \{1, \dots, n\}^2$ represents the edge set with the the the $n$ nodes in $G_p$, and $A_p \in \mathbb{R}^{|E_p| \times d_e}$ contains the edge attributes corresponding to $E_p$. The example of the patient subgraph is illustrated Figure 2.

### 3.2.1 MODEL

**GNN module** For each $G_p$, we apply a multi-layer graph convolutional network encoder to refine the original node embeddings. Let $H^{(0)} = X_p$ denote the initial node features. Node representations are updated across $L$ layers as follows:

$$H^{(l)} = \text{Dropout}\left(\sigma\left(\text{LayerNorm}\left(\text{GATv2}(H^{(l-1)}, E, A)\right)\right)\right), \quad l = 1, \dots, L-1,$$

where $\sigma(\cdot)$ is a non-linear activation function, $\text{LayerNorm}(\cdot)$ denotes layer normalization, and $\text{Dropout}(\cdot)$ is a dropout operation applied for regularization during training. The final node representations are projected via a learnable linear map ($f_{\text{proj}}$):

$$Z = f_{\text{proj}}\left(\text{GATv2}(H^{(L-1)}, E, A)\right).$$

The GATv2 is an attention graph operator from (Brody et al., 2022).

Let $\{z_i\}_{i=1}^n$ be the node embeddings output by the GNN module. We partition these into phenotype and gene sets.

**Patient representation learning module** Phenotype embeddings are first projected by a multi-layer perceptron (MLP), denoted by $f_{\text{pheno}}$:

$$\tilde{z}_i = f_{\text{pheno}}(\{z_i\}_{i \in \mathcal{P}_p}).$$

To incorporate global context, $m$ learnable memory vectors $M \in \mathbb{R}^{m \times d}$ are concatenated with the phenotype embeddings and passed through multi-head self-attention:

$$\hat{Z} = \text{MHA}([\tilde{Z}; M]),$$

where MHA denotes standard multi-head attention as introduced in (Vaswani et al., 2017).

The patient representation $p \in \mathbb{R}^d$ is then obtained by masked mean pooling over phenotype nodes, followed by a two-layer MLP denoted by $f_{\text{patient}}$:

$$p = f_{\text{patient}}\left(\frac{1}{|\mathcal{P}_p|} \sum_{i \in \mathcal{P}_p} \hat{Z}_i\right).$$

**Gene Encoder** Gene node embeddings are processed via a Transformer-based encoder (Vaswani et al., 2017), parameterized by $\theta$:

$$\mathbf{G} = \theta(\{z_j\}_{j \in \mathcal{G}_p}),$$

where $\mathcal{G}_p$ are the gene indices in the patient graph and $\mathbf{G} \in \mathbb{R}^{L \times d}$ are gene embeddings.

### 3.3 LOSSES

**Gene Loss** To train the model, we adopt a contrastive loss framework. Let $p \in \mathbb{R}^d$ be the patient embedding, $\mathbf{G} = [g_1, \dots, g_L] \in \mathbb{R}^{L \times d}$ the candidate gene embeddings, and $g^*$ the embedding of the causative gene. All embeddings are normalized to unit norm. We compute cosine similarities, where $\tau$ is a learnable temperature parameter:

$$\text{sim}(p, g_i) = \frac{\langle p, g_i \rangle}{\tau},$$

Let $\text{sim}^* = \text{sim}(p, g^*)$ be the similarity to the causative gene. We apply semi-hard negative mining by first selecting a candidate set of negative genes satisfying:

$$\text{sim}(p, g^-) < \text{sim}^* - \gamma,$$

where $\gamma$ is a margin hyperparameter. Among this candidate set, we select the negative gene with the highest similarity (i.e., the hardest semi-hard negative). If no semi-hard candidate exists, we select the negative gene with the highest overall similarity. This strategy balances learning from informative negatives while avoiding extreme outliers.

The loss is then computed using a margin-based triplet loss:

$$\mathcal{L}_{\text{triplet}} = \max(0, \text{sim}(p, g^-) - \text{sim}(p, g^*) + \gamma).$$

We further add an L2 norm regularization term to encourage both patient and gene embeddings to remain close to unit norm:

$$\mathcal{L}_{\text{reg}} = \lambda \left| \|p\|_2 + \text{mean}\left(\|\mathbf{G}\|_2\right) - 2 \right|,$$

where $\lambda$ is a regularization weight and $\|\mathbf{G}\|_2$ is the mean L2 norm across candidate gene embeddings.

The total loss is:

$$\mathcal{L}_{\text{gene}} = \mathcal{L}_{\text{triplet}} + \mathcal{L}_{\text{reg}}.$$

**Patient Similarity Loss**    To encourage consistency across batches and improve generalization, we maintain a memory bank $\mathcal{M}$ of past patient embeddings and their associated gene labels. Let $\mathcal{M} = \{(p_i, g_i)\}$ be the set of embeddings in the memory bank.

The patient similarity loss is computed based on cosine similarities between patient embeddings having the same causative gene in the current and previous batches. We define:

$$\mathcal{L}_{\text{sim}} = \mathcal{L}_{\text{within}} + \mathcal{L}_{\text{cross}},$$

where $\mathcal{L}_{\text{within}}$ compares embeddings within the current batch, and $\mathcal{L}_{\text{cross}}$ compares them against the memory bank.

Each of $\mathcal{L}_{\text{within}}$ and $\mathcal{L}_{\text{cross}}$ consists of two parts: pulling same-gene pairs together and pushing different-gene pairs apart.

$$\mathcal{L}_{\text{pull}} = -\log \sigma \left( \frac{\text{sim}(p_i, p_j)}{\alpha} \right), \quad \text{for } g_i = g_j$$

$$\mathcal{L}_{\text{push}} = \max(0, \delta - (1 - \text{sim}(p_i, p_j))) \quad \text{for } g_i \neq g_j$$

The final loss becomes:

$$\mathcal{L}_{\text{total}} = \mathcal{L}_{\text{gene}} + \mathcal{L}_{\text{sim}},$$

with $\delta$ the margin and $\alpha$ the temperature. The memory bank is updated per batch using a circular buffer strategy to retain a fixed number of embeddings.

### 3.4    INFERENCE

At inference time, we construct the patient-specific subgraph $G_p$ by extracting the $k$-hop neighborhood around each phenotype in $\mathcal{P}_p$, thereby capturing the set of candidate genes $\mathcal{G}_p$ associated with the patient's clinical profile (Section 3.2). The model encodes $G_p$ to produce a patient embedding $p$ and gene embeddings $\mathbf{G} = \{g_j\}_{j \in \mathcal{G}_p}$.

Relevance scores are computed via:

$$s^{\text{orig}}(g) = \text{sim}(p, g_j) = \langle p, g_j \rangle, \quad \forall j \in \mathcal{G}_p.$$

The final output is a ranked list of candidate genes:

$$\mathcal{G}_p^{\text{ranked}} = \text{argsort}_{j \in \mathcal{G}_p} \left( \text{sim}(p, g_j) \right).$$

**Reranking Procedure.** To incorporate up-to-date medical knowledge, we rerank predictions by combining scores from two sources: (i) the *original model-based score* $s^{\mathrm{orig}}(g)$, and (ii) the *phenotype-based score* $s^{\mathrm{pheno}}(g)$.

The phenotype-based score is defined as the overlap between the patient's phenotype set $\mathcal{P}_p$ and the phenotype list associated with gene $g$ from external databases, like Monarch for example (Putman et al., 2024). Formally, each gene $g$ from the original set of candidate genes $\mathcal{G}_p$ has associations in the external databases

$$g \mapsto [\mathrm{phenotypes}(g)].$$

The phenotype score is given by

$$s^{\mathrm{pheno}}(g) \propto \big| \mathcal{P}_p \cap \mathrm{phenotypes}(g) \big|.$$

Both scores are normalized to $[0, 1]$. The final combined score is

$$s^{\mathrm{comb}}(g) = \hat{\alpha} \cdot \tilde{s}^{\mathrm{pheno}}(g) + \hat{\beta} \cdot \tilde{s}^{\mathrm{orig}}(g), \quad \hat{\alpha} + \hat{\beta} = 1.$$

Genes in the patient subgraph are then reranked in descending order of $s^{\mathrm{comb}}(g)$. This addresses a common limitation of knowledge graphs, which are typically fixed and may not reflect the most recent medical knowledge. By integrating phenotype-driven evidence with similarity-based model scores, the reranking step incorporates up-to-date knowledge into the predictions.

## 4 EXPERIMENTS AND RESULTS

### 4.1 DATASET

Our method leverages a biomedical knowledge graph (KG) to augment patient information and identify candidate genes. We use PrimeKG (Chandak et al., 2023), adapted for rare disease tasks in (Alsentzer et al., 2025), comprising 105,220 nodes and 1,095,469 edges across seven entity types (phenotypes, diseases, genes/proteins, pathways, molecular functions, cellular components, and biological processes) defined by vocabularies such as HPO (Köhler et al., 2019) and EN-SEMBL (Aken et al., 2016). Relations span 17 edge types, including protein–protein interactions (321,075), disease–phenotype (204,779 positive, 1,483 negative), phenotype–phenotype (21,925), phenotype–protein (10,518), and disease–protein associations (86,299). Full KG specifications are available in (Chandak et al., 2023; Alsentzer et al., 2025).

We used three real-world datasets and one simulated cohort in our experiments. The model was trained on a simulated dataset from (Alsentzer et al., 2023), which includes 36,224 patients for training and 6,080 patients for validation. Simulated data was chosen for its similarity to the Undiagnosed Diseases Network (UDN) dataset (Ramoni et al., 2017), its larger size for training complex models, and its public availability. Each simulated patient includes positive phenotypes and a challenging candidate gene list. Patient subgraphs for the training and validation sets were constructed by identifying the shortest paths between each phenotype and each candidate gene and incorporating all intermediate nodes to preserve the underlying biological relationships.

For evaluation, we used a random subset of 320 patients from the simulated dataset as the final test set and real-world datasets. Simulated dataset considered in two settings: with a candidate gene list, comprising all 320 patients, and without, comprising 306 patients after preprocessing. For real-world data, we included several benchmark datasets: the Facial Phenotype Gene Disease dataset (FGDD) with 738 patients (Song et al., 2025), PhenoDis with 109 patients (Adler et al., 2018), and MyGene2 with 135 patients (University of Washington, Center for Mendelian Genomics). PhenoDis is a comprehensive resource for the phenotypic characterization of rare cardiac diseases. For it, we retrieved causal genes via the Monarch API using disease names. For FGDD and PhenoDis, we kept only patients with a single causative gene, at least one phenotype, and removed duplicates or empty records. Detailed preprocessing steps can be found in Appendix. For MyGene2 data collection and preprocessing followed (Alsentzer et al., 2025). While MyGene2 is a commonly used benchmark, its gene distribution is highly imbalanced, which can inflate performance estimates (see Appendix for details). Since none of the real-world datasets provide expert-curated candidate gene lists, they were used to assess model performance in the setting without candidates.

Table 1: Final dataset characteristics after preprocessing.

| Dataset | Patients | Phenotypes / Patient | Unique Phenotypes | Unique Genes |
|---|---|---|---|---|
| Training Dataset | 36,224 | 18.5 ± 7.7 | 6,152 | 2,164 |
| FGDD | 738 | 7.3 ± 4.0 | 369 | 213 |
| Simulated (full) | 320 | 17.8 ± 7.0 | 2,152 | 208 |
| Simulated (2-hop valid) | 306 | 18.0 ± 7.0 | 2,109 | 200 |
| MyGene2 | 135 | 8.3 ± 6.7 | 288 | 42 |
| PhenoDis | 109 | 30.4 ± 36.3 | 1,327 | 86 |
| **All Test Sets** | **1,288** | - | **3,021** | **514** |

For computational efficiency, evaluation on PrimeKG was limited to the 2-hop neighborhood around each patient's phenotypes, which preserves clinically plausible gene candidates while preventing the exponential growth observed at 3 hops (Appendix). Patients whose causal gene was not present in this 2-hop region were excluded from all datasets. To reduce noise from weak or incidental phenotype–gene links, we further applied soft filtering, retaining only genes that appeared in 25% of a patient's subgraphs. Importantly, hop depth and filtering are not a constraint of the framework itself, PhenoKG can operate with k-hop graphs, predefined genetic panels, or without filtering, when appropriate. All patients whose causal gene was removed during filtering were excluded from training and evaluation to avoid artificially injecting ground truth signals. Final dataset statistics are reported in Table 1, which shows substantial differences across datasets (e.g., dense phenotype profiles in PhenoDis vs. sparse annotations in FGDD/MyGene2). Full details of each step of datasets preprocessing, details discussion on subgraphs creation and filtering as well as dataset overlap analyses are provided in the Appendix.

For reranking, we incorporated knowledge from the Monarch database (Putman et al., 2024) by retrieving, for each phenotype in the test datasets, the list of associated causal genes with on average 119 genes per phenotype and mapping them to gene–phenotype associations. This filtering step resulted in a mean gene list of 1,828 genes per patient. For MyGene2 we also extended the associations map with HPO annotation dataset Köhler et al. (2021). Worth noting is reranking is optional and any additional source can be used for that.

We use the terms *gene* and *protein* in this section interchangeably, as genes are represented by their encoded proteins in the knowledge graph. This reflects that most biological interaction data, such as protein-protein interactions and functional annotations, are defined at the protein level. Thus, gene references correspond to protein nodes acting as proxies for genes.

## 4.2 EXPERIMENTAL SETUP

We benchmark our method against Amelie (Birgmeier et al., 2020), PhenoApt (Chen et al., 2022) and Phen2Gene Zhao et al. (2020), and include Shepherd as the most structurally comparable KG-based approach. For Shepherd we relied on publicly released checkpoints and for Amelie, PhenoApt and Phen2Gene on API access, and therefore report single runs without standard deviations. We do not evaluate Phenomizer (Ullah et al., 2013), Phenolyzer (Yang et al., 2015), or Phrank (Jagadeesh et al., 2019), as PhenoApt has already been shown to outperform them in settings without candidate gene lists. Similarly, we exclude LIRICAL (Robinson et al., 2020), ERIC (XRare) (Li et al., 2019), CADA, HiPhive (Exomiser) (Smedley et al., 2015), and PhenIX (Exomiser) (Smedley et al., 2015), since Shepherd has previously demonstrated superior performance over these methods. In addition, we evaluated GPT-4o (Achiam et al., 2023) via the OpenAI API as a large language model baseline by prompting it to rank candidate genes directly from patient phenotypes and candidate lists, using a structured JSON output format. For Shepherd, we used the same candidate list retrieved from the $k$-hop neighborhood as for our model. PhenoApt and Phen2Gene do not require a candidate gene list. Variability is reported via 3 independent runs with different seeds, not hypothesis testing, since PhenoKG outputs embedding-space similarity ranking, not parametric statistics

We evaluated our model using MRR, Mean Rank, and Hits@$j$. MRR measures average ranking quality, Mean Rank the average position of the causative gene (lower is better), and Hits@$j$ the

Table 2: Performance comparison of Shepherd, PhenoApt, Phen2Gene and Ours across four datasets (PhenoDis, FGDD, MyGene2, Simulated without candidate genes list). Metrics are percentages (mean ± std for Ours across 3 separate runs). Bold values denote the best result per dataset and column.

| Dataset / Model | MRR | Hits@1 | Hits@3 | Hits@5 | Hits@10 | Hits@15 | Hits@20 | Hits@25 | Hits@30 | Hits@50 | Hits@75 | Hits@100 |
|---|---|---|---|---|---|---|---|---|---|---|---|---|
| **PhenoDis** | | | | | | | | | | | | |
| Shepherd | 22.2 | 13.8 | 18.3 | 26.6 | 39.4 | 55.0 | 59.6 | 62.4 | 69.7 | 87.2 | 90.8 | 95.4 |
| Phen2Gene | 34.7 | 25.7 | 35.8 | 45.0 | 56.9 | 57.8 | 63.3 | 66.1 | 67.9 | 82.6 | 88.1 | 91.7 |
| PhenoApt | 43.9 | 30.3 | 49.5 | 63.3 | 73.4 | 78.0 | 78.9 | 79.8 | 82.6 | 87.2 | 89.9 | 91.7 |
| Ours | **62.3 ± 3.3** | **50.8 ± 6.6** | **69.4 ± 1.4** | **79.8 ± 1.6** | **85.6 ± 0.5** | **86.2 ± 0.0** | **87.8 ± 1.1** | **89.0 ± 0.9** | **90.5 ± 0.5** | **92.7 ± 0.9** | **93.6 ± 0.9** | **94.2 ± 1.1** |
| **FGDD** | | | | | | | | | | | | |
| Shepherd | 11.4 | 6.4 | 11.1 | 14.4 | 21.4 | 24.4 | 30.1 | 33.2 | 35.9 | 43.8 | 50.5 | 56.1 |
| Phen2Gene | 16.1 | 10.0 | 18.7 | 21.8 | 27.4 | 32.3 | 33.7 | 35.4 | 36.7 | 42.0 | 49.5 | 54.3 |
| PhenoApt | 25.3 | 16.7 | **28.9** | 33.1 | 39.0 | 42.7 | 46.9 | 49.3 | 52.0 | 56.9 | 63.6 | 68.7 |
| Ours | **25.7 ± 1.0** | **17.0 ± 1.6** | 27.8 ± 0.1 | **34.3 ± 0.5** | **44.3 ± 0.6** | **48.6 ± 1.8** | **53.2 ± 0.2** | **55.5 ± 1.0** | **57.7 ± 0.5** | **64.8 ± 0.7** | **70.9 ± 0.3** | **75.1 ± 0.2** |
| **MyGene2** | | | | | | | | | | | | |
| Shepherd | 17.1 | 10.4 | 16.3 | 20.7 | 25.9 | 44.4 | 53.3 | 54.8 | 57.8 | 62.2 | 66.7 | 77.8 |
| Phen2Gene | 12.5 | 9.6 | 11.9 | 14.8 | 17.8 | 20.0 | 21.5 | 25.2 | 26.7 | 40.0 | 47.4 | 49.6 |
| PhenoApt | **56.0** | **45.9** | **65.9** | **68.9** | 74.1 | 76.3 | 77.8 | 79.3 | 80.7 | 83.7 | 88.1 | 89.6 |
| Ours | 50.9 ± 3.7 | 40.7 ± 5.2 | 55.6 ± 3.4 | 63.2 ± 5.2 | **74.8 ± 0.7** | **77.3 ± 0.9** | **78.3 ± 1.1** | 79.0 ± 0.4 | 79.8 ± 0.4 | **84.2 ± 0.9** | 87.7 ± 1.1 | **89.6 ± 0.0** |
| **Simulated** | | | | | | | | | | | | |
| Shepherd | 17.1 | 10.4 | 16.3 | 20.7 | 25.9 | 44.4 | 53.3 | 54.8 | 57.0 | 61.0 | 68.0 | 71.0 |
| Pheno2Gene | 26.7 | 16.9 | 28.8 | 38.6 | 49.0 | 54.3 | 58.8 | **62.1** | **64.1** | **72.6** | **77.1** | 79.4 |
| PhenoApt | 25.3 | 16.7 | 28.9 | 33.1 | 39.0 | 42.7 | 46.9 | 49.3 | 52.0 | 56.9 | 63.6 | 68.7 |
| Ours | **32.0 ± 1.9** | **22.5 ± 2.5** | **35.3 ± 1.3** | **42.6 ± 1.0** | **50.5 ± 1.8** | **54.8 ± 2.0** | **58.8 ± 0.9** | 61.1 ± 0.9 | 63.3 ± 1.2 | 70.8 ± 1.9 | 76.3 ± 0.2 | **80.6 ± 0.5** |

Table 3: Results on the Simulated dataset with candidate genes. Metrics reported: MRR, Mean Rank, and Hits@K (K=1,3,5,10). Values are means; for Ours and GPT-4o, mean ± std across 3 separate runs is reported.

| Model | MRR (%) | Mean Rank | Hits@1 (%) | Hits@3 (%) | Hits@5 (%) | Hits@10 (%) |
|---|---|---|---|---|---|---|
| Shepherd | 78.6 | 1.82 | 65.9 | 88.4 | 95.6 | 99.4 |
| PhenoApt | 11.8 | 65.48 | 7.2 | 10.9 | 14.7 | 20.6 |
| Amelie | 68.0 | 2.41 | 51.8 | 80.2 | 89.5 | 99.0 |
| GPT-4o | 24.3 ± 0.9 | 7.54 ± 0.12 | 8.0 ± 0.6 | 23.8 ± 1.6 | 37.2 ± 1.5 | 73.4 ± 0.3 |
| Ours | **89.2 ± 0.8** | **1.575 ± 0.005** | **83.9 ± 1.2** | **92.8 ± 0.5** | **96.2 ± 0.6** | **98.5 ± 0.5** |

success rate within the top-$j$. Mean Rank is reported only in the candidate gene setting, since methods like PhenoApt truncate outputs at 100, making it uninformative otherwise.

## 4.3 IMPLEMENTATION DETAILS

We pre-trained the global graph $G$ on a link prediction task (Alsentzer et al., 2025) with 512-dimensional output embeddings. The GNN module used $L = 3$ GATv2 (Brody et al., 2022) layers with $h = 2$ attention heads per layer, hidden dimensions $d_{\text{hid}}^{(1)} = 1024$, $d_{\text{hid}}^{(2)} = 256$, and output $d_{\text{out}} = 512$. LayerNorm (Ba et al., 2016) and LeakyReLU (Maas et al., 2013) were applied between layers, with dropout $p = 0.4$. Edge attributes ($d_e = 15$) were incorporated into the attention mechanism to enhance relational modeling, and a final linear transformation was applied to project the resulting node representations into the output embedding space.

For patient representation learning, phenotype projection $f_{\text{pheno}}$ used a two-layer MLP with ReLU (Krizhevsky et al., 2012) and LeakyReLU. The memory bank contained $m = 128$ learnable vectors (normal init., mean 0, std 1). Phenotype attention used 4 attention heads. Patient embeddings were aggregated and passed through a two-layer MLP with LeakyReLU. Gene embeddings were processed by a transformer encoder (Vaswani et al., 2017) with 4 layers, 8 attention heads, and an intermediate size of 2048. Model training was performed using the AdamW optimizer (Loshchilov & Hutter, 2017) with a learning rate of $1 \times 10^{-4}$, combined with a cosine annealing learning rate scheduler (Loshchilov & Hutter, 2016) configured with $T_0 = 10$ and a multiplier $T_{\text{mult}} = 2$. The patient neighbourhood was defined using $k = 2$ nearest neighbours. The following hyperparameters were used: margin $\gamma = 0.3$, contrastive temperature $\tau = 0.12$, regularization weight $\lambda = 0.03$, patient similarity temperature $\alpha = 0.5$, and similarity margin $\delta = 0.8$. We set $\hat{\alpha} = 0.7$ and $\hat{\beta} = 0.3$ in our experiments. These values were selected as fixed hyperparameters (see Appendix). All models were implemented and trained using PyTorch, PyTorch Geometric, and Transformers. Encoder training ran for up to 135 epochs with early stopping after 25 epochs.

## 4.4 RESULTS AND DISCUSSIONS

We evaluated our method in two settings: (i) without candidate gene lists, comparing against Shepherd and PhenoApt (Table 2), and (ii) with candidate gene lists, comparing against Shepherd, Amelie, PhenoApt, and GPT-4o (Table 3).

**Without candidate gene lists.** Our model consistently outperformed Shepherd and matched or exceeded PhenoApt on most datasets. On PhenoDis, our method achieved the best results overall with an MRR of 62.3%, surpassing both Shepherd (22.2%) and PhenoApt (43.9%). On FGDD, we reached an MRR of 25.7%, slightly higher than PhenoApt (25.3%) and more than double Shepherd (11.4%). While the MRR values for our method and PhenoApt are close on FGDD, our approach delivered stronger recall across the ranking (e.g. Hits@10 of 44.3% vs. 39.0% for PhenoApt, and Hits@100 of 75.1% vs. 68.7%). On MyGene2, we improved MRR from 17.1% (Shepherd) to 31.1% and Hits@100 from 77.8% to 85.4%, although PhenoApt remained ahead with 56.0% MRR. On the simulated dataset, which is particularly challenging due to noisy phenotype assignments, our model again outperformed both Shepherd (17.1% MRR) and PhenoApt (25.3% MRR), reaching 32.0% MRR.

**With candidate gene lists.** Our method achieved clear state-of-the-art performance with an MRR of 89.2% and Hits@1 of 83.9%, outperforming Shepherd (78.6% MRR, 65.9% Hits@1), Amelie (68.0% MRR, 51.8% Hits@1), and GPT-4o (24.8% MRR, 8.7% Hits@1). PhenoApt, which cannot incorporate candidate lists, showed substantially lower performance in this setting (11.8% MRR).

We performed extensive ablation studies to assess the contribution of each component of our framework. These experiments disentangle the effect of loss functions, embedding initialization, and input graph design, showing that the combined loss and pretrained embeddings consistently yield the strongest performance as well as ablation experiments on different level of filtering. Full quantitative results and additional analyses are provided in Appendix.

## 5 LIMITATIONS

Our method cannot predict genes absent from the knowledge graph; this occurred only several times across all datasets. However, if new nodes, like genes or phenotypes are connected to existing ones, embeddings can be extrapolated and predictions remain possible. Also, shortest-path subgraph extraction at 3+ hops is resource intensive since it means computing the shortest paths between each phenotype and 16-20k genes.

## 6 CONCLUSION

We introduced PhenoKG, a phenotype-driven knowledge graph framework that prioritizes causative genes directly from patient phenotypes. PhenoKG achieves robust performance across diverse benchmarks, outperforming state-of-the-art methods on multiple rare disease benchmarks. It demonstrates strong performance in both candidate and non-candidate settings, reflecting realistic diagnostic conditions. The model is best positioned as a pre-filter in diagnostic pipelines, narrowing the search space before variant analysis. Future work will focus on dynamic graph updates and multimodal extensions to improve generalization and clinical applicability.

## 7 REPRODUCIBILITY STATEMENT

We have taken several steps to ensure the reproducibility of our results. All datasets used in this work are explicitly limited to publicly available resources, and we describe them in detail in the main paper. The full set of implementation details, hyperparameters, and training procedures are provided in Section 4 and the Appendix, ensuring that our experiments can be replicated. Upon acceptance, we will release the source code including all the files to preprocess the data and configuration files to enable exact reproduction of all reported results.

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

# A    APPENDIX

Table 4: Evaluation results using Mean Rank, MRR, and Hits@J (in %) are reported across different model groups for the MyGene2 dataset. Lower values indicate better performance for Mean Rank, while higher values indicate better performance for MRR and Hits@J. The *Pretrained embeddings* group corresponds to training initialized with embeddings pretrained on the knowledge graph for a link prediction task. The *No embeddings* group refers to training from scratch. The *Only patient phenotypes* group includes models trained solely on original patient phenotype data. Rows within each group represent different training setups using various loss function combinations.

| Model | | Mean Rank↓ | MRR↑ | Hits@1↑ | Hits@3↑ | Hits@5↑ | Hits@10↑ | Hits@25↑ | Hits@50↑ | Hits@75↑ | Hits@100↑ | Hits@150↑ | Hits@200↑ | Hits@250↑ |
|---|---|---|---|---|---|---|---|---|---|---|---|---|---|---|
| **PhenoKG (Ours)** | | | | | | | | | | | | | | |
| $\mathcal{L}_{\text{gene}}$ | $\mathcal{L}_{\text{sim}}$ | | | | | | | | | | | | | |
| *No Embeddings* | | | | | | | | | | | | | | |
| ✓ | ✗ | 302.94±35.92 | 9.69±0.67 | 3.46±0.92 | 9.38±1.26 | 16.05±3.04 | 20.74±3.68 | 31.36±1.85 | 42.22±3.97 | 48.40±6.87 | 57.04±6.87 | 64.69±6.98 | 71.11±7.13 | 77.78±5.96 |
| ✗ | ✓ | 363.50±27.34 | 7.62±2.42 | 1.48±0.60 | 9.14±4.46 | 11.11±4.80 | 15.56±5.77 | 28.64±3.54 | 40.00±1.21 | 46.67±2.42 | 52.10±3.04 | 62.47±4.29 | 70.12±2.52 | 76.05±1.26 |
| ✓ | ✓ | 487.80±95.32 | 11.24±1.23 | 6.67±2.18 | 10.86±1.52 | 17.53±3.70 | 21.23±3.54 | 26.42±3.10 | 37.04±4.23 | 44.20±5.14 | 48.40±4.03 | 56.79±1.26 | 61.48±0.00 | 66.42±3.04 |
| *Only Patient Phenotypes* | | | | | | | | | | | | | | |
| ✓ | ✗ | 232.38±17.08 | 19.45±0.94 | 13.09±1.26 | 21.73±0.35 | 24.94±0.70 | 29.14±1.52 | 40.25±1.52 | 58.77±0.92 | 65.68±2.44 | 68.89±2.10 | 75.06±3.33 | 79.75±0.70 | 81.98±0.35 |
| ✗ | ✓ | 451.99±115.66 | 15.80±2.97 | 4.94±2.79 | 20.00±4.36 | 26.42±2.12 | 36.30±6.31 | 52.10±4.03 | 60.99±4.46 | 62.96±4.36 | 68.15±3.97 | 72.59±4.23 | 75.31±4.12 | 77.28±4.25 |
| ✓ | ✓ | 364.78±110.58 | 17.47±1.00 | 6.17±1.52 | 23.70±1.81 | 28.64±0.35 | 32.84±1.94 | 47.65±5.21 | 67.65±1.75 | 70.62±2.86 | 74.07±2.64 | 75.06±3.33 | 77.28±3.33 | 81.23±3.10 |
| *Pretrained Embeddings* | | | | | | | | | | | | | | |
| ✓ | ✗ | 202.49±12.63 | 19.10±1.32 | 11.36±3.33 | 21.48±0.60 | 26.67±2.64 | 30.86±3.04 | 43.21±11.67 | 57.53±5.62 | 62.96±4.36 | 70.62±3.33 | 77.78±0.60 | 80.49±0.35 | 82.47±0.70 |
| ✗ | ✓ | 315.57±27.41 | 18.88±0.52 | 9.38±1.94 | 21.98±2.44 | 28.15±3.02 | 33.33±5.24 | 52.10±3.33 | 65.68±1.26 | 68.89±2.77 | 70.86±2.44 | 74.57±2.52 | 77.53±3.04 | 79.75±1.52 |
| ✓ | ✓ | 333.44±83.36 | 19.32±3.04 | 11.11±2.18 | 21.98±3.84 | 25.43±3.94 | 32.10±5.59 | 49.38±10.43 | 59.26±4.95 | 63.70±4.36 | 68.89±3.37 | 72.84±2.29 | 74.81±2.10 | 77.04±2.18 |
| *Filtered* | | | | | | | | | | | | | | |
| ✓ | ✓ | 112.27±13.25 | 50.9±3.7 | 40.7±5.2 | 55.6±3.4 | 63.2±5.2 | 74.8±0.7 | 79.0±0.4 | 84.2±0.9 | 86.6±1.0 | 89.6±0.0 | 91.1±1.0 | 91.8±0.9 | 92.5±0.8 |

Table 5: Performance of dataset across different $\alpha$ and $\beta$ weighting schemes. Reported as mean $\pm$ std.

| $\alpha$ | $\beta$ | MRR | Mean Rank | Hits@1 | Hits@5 | Hits@10 | Hits@25 | Hits@50 | Hits@100 |
|---|---|---|---|---|---|---|---|---|---|
| 0.0 | 1.0 | 0.193 ± 0.037 | 333.437 ± 102.099 | 0.111 ± 0.027 | 0.254 ± 0.048 | 0.321 ± 0.068 | 0.494 ± 0.128 | 0.593 ± 0.061 | 0.689 ± 0.041 |
| 0.1 | 0.9 | 0.213 ± 0.038 | 131.521 ± 15.368 | 0.116 ± 0.030 | 0.267 ± 0.034 | 0.358 ± 0.101 | 0.615 ± 0.116 | 0.758 ± 0.048 | 0.849 ± 0.015 |
| 0.2 | 0.8 | 0.225 ± 0.039 | 123.640 ± 14.156 | 0.116 ± 0.030 | 0.286 ± 0.034 | 0.415 ± 0.149 | 0.694 ± 0.076 | 0.805 ± 0.011 | 0.869 ± 0.015 |
| 0.3 | 0.7 | 0.249 ± 0.038 | 119.795 ± 13.631 | 0.128 ± 0.024 | 0.338 ± 0.086 | 0.506 ± 0.160 | 0.753 ± 0.019 | 0.815 ± 0.000 | 0.886 ± 0.004 |
| 0.4 | 0.6 | 0.283 ± 0.052 | 117.356 ± 13.474 | 0.158 ± 0.050 | 0.415 ± 0.115 | 0.625 ± 0.089 | 0.793 ± 0.007 | 0.817 ± 0.004 | 0.894 ± 0.004 |
| 0.5 | 0.5 | 0.335 ± 0.065 | 115.101 ± 13.090 | 0.207 ± 0.071 | 0.511 ± 0.122 | 0.689 ± 0.034 | 0.798 ± 0.004 | 0.832 ± 0.004 | 0.894 ± 0.004 |
| 0.6 | 0.4 | 0.407 ± 0.058 | 113.521 ± 13.191 | 0.267 ± 0.068 | 0.583 ± 0.070 | 0.731 ± 0.004 | 0.795 ± 0.000 | 0.837 ± 0.007 | 0.894 ± 0.004 |
| 0.7 | 0.3 | 0.509 ± 0.037 | 112.272 ± 13.247 | 0.407 ± 0.052 | 0.632 ± 0.052 | 0.748 ± 0.007 | 0.790 ± 0.004 | 0.842 ± 0.009 | 0.896 ± 0.000 |
| 0.8 | 0.2 | 0.544 ± 0.040 | 112.005 ± 13.782 | 0.449 ± 0.059 | 0.635 ± 0.024 | 0.756 ± 0.007 | 0.788 ± 0.011 | 0.837 ± 0.007 | 0.891 ± 0.004 |
| 0.9 | 0.1 | 0.552 ± 0.038 | 111.706 ± 13.291 | 0.464 ± 0.048 | 0.644 ± 0.020 | 0.753 ± 0.011 | 0.783 ± 0.004 | 0.835 ± 0.009 | 0.884 ± 0.004 |
| 1.0 | 0.0 | 0.368 ± 0.000 | 118.891 ± 12.642 | 0.281 ± 0.000 | 0.452 ± 0.000 | 0.600 ± 0.000 | 0.719 ± 0.000 | 0.793 ± 0.000 | 0.874 ± 0.000 |

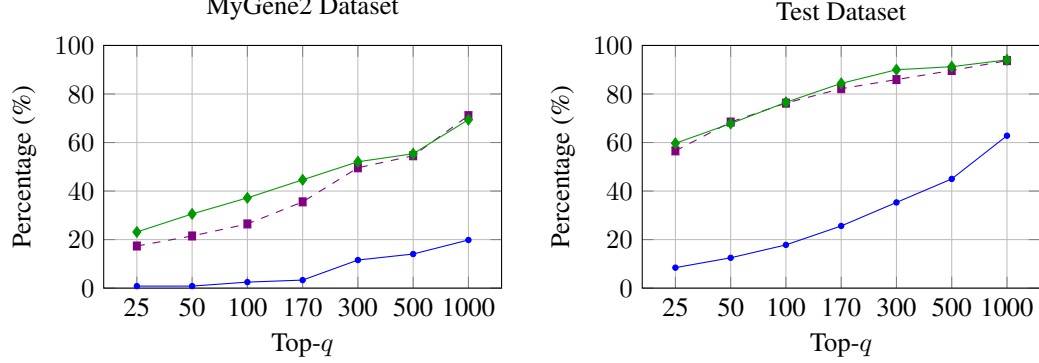

Figure 3: Match percentage comparison for MyGene2 (left) and simulated test (right) datasets, across top-$q$ values. Blue solid line with circles: PhenoKG with $\mathcal{L}_{\text{gene}}$; violet dashed line with squares: with combined $\mathcal{L}$ loss ; green line with diamonds: model with $\tilde{\mathcal{L}}_{\text{sim}}$.

## A.1    ABLATION STUDIES

**Loss functions.**    We compared models trained with gene loss only, similarity loss only, and a combined loss. Using $\mathcal{L}_{\text{gene}}$ alone produced competitive gene prioritization results but weak patient–patient alignment, as reflected in poor match percentages in Figure 3. Models trained with $\mathcal{L}_{\text{sim}}$

alone achieved strong patient similarity retrieval but underperformed on direct causative gene ranking. The combined loss provided a balanced trade-off, maintaining good performance on gene prioritization while enabling extension to tasks such as patient similarity, which were not evaluated here. Notably, using only gene loss produced higher scores at larger cutoffs (e.g., Hits@75), indicating that if the sole objective is gene prioritization or generating candidate lists within the top 150 ranked genes, training with gene loss alone may be preferable (Table 4).

**Embedding initialization.** We investigated two embedding strategies: (i) *Pretrained embeddings*, initialized from link prediction pretraining; (ii) *No embeddings*, trained from scratch; Pretraining improved generalization on MyGene2, which contains novel phenotype–gene pairs absent during training. Pretrained embeddings yielded higher performance on smaller cutoffs (e.g., Hits@1–10). When training and test data fully overlap, pretraining is not required, but it is crucial for generalizing to unseen genes and phenotypes, which is the case for most of our test datasets.

**Only patient phenotypes.** All our models were trained under a realistic setting where patient phenotypes were made noisy, so patient embeddings incorporated not only the true phenotypes but also additional nodes from the patient graphs. This design improves generalization to real-world scenarios where phenotype data are inherently noisy. We tested the *Only patient phenotypes* variant, which restricts graphs to real patient nodes, and found that its performance was overall unstable.

**Patient similarity analysis.** As shown in Figure 3, the combined loss markedly increased the proportion of patients whose nearest neighbors shared the same causative gene. On MyGene2, match percentages improved from under 20% (gene loss only) to over 70% (combined loss). On simulated data, alignment improved even further, reflecting that embedding patients in a shared space captures clinically meaningful relationships.

## A.2 DATASETS

### A.2.1 FGDD AND PHENODIS CLEANING

We evaluate our method on two complementary rare-disease datasets: the clinical FGDD cohort and the knowledge-base–derived PhenoDis. Both require normalization into a monogenic, patient-level format compatible with gene-ranking evaluation. The complete filtering pipeline for each dataset is summarized in Table 6.

**FGDD (clinical dataset).** FGDD contains real patient cases with assigned causal genes and recorded HPO terms. Preprocessing focuses on ensuring valid supervision and removing structural inconsistencies. We exclude entries without known causal genes, map all gene symbols to standardized Ensembl identifiers, remove records with multiple causal genes, deduplicate identical patient entries, and discard cases without positive phenotype annotations. These steps produce a clean, single-gene patient cohort with consistent gene identifiers and reliable phenotype profiles.

**PhenoDis (knowledge-base–derived dataset).** PhenoDis originates from disease-level Orphanet entries and must be converted into synthetic patient records. The main challenge is mapping diseases to causal genes; diseases with no known gene associations are removed. We then construct patient profiles from the corresponding HPO terms, enforce monogenic supervision by excluding diseases associated with multiple genes, and normalize phenotype identifiers for compatibility with downstream tools. This pipeline yields richly phenotyped synthetic patients aligned with a single-gene supervision objective.

### A.2.2 PROBLEMS WITH THE MYGENE2 DATASET

While MyGene2 is a widely used benchmark, its gene distribution is highly skewed: four genes (PIEZO2, NALCN, SF3B4, and MYH3) account for 63% of patinets in the test set. This concentration tends to inflate performance for methods that exploit frequency patterns. For example, PhenoApt achieves an MRR of 89.6% on PIEZO2 (36 patients) and 90.4% on NALCN (15 patients), whereas its MRR for NALCN in the simulated cohort drops to only 0.6%, reflecting the more challenging phenotype structure in the simulated dataset.

Table 6: Preprocessing pipeline comparison for FGDD and PhenoDis datasets.

| Processing Step | FGDD | | | PhenoDis | | |
|---|---|---|---|---|---|---|
| | Remaining | Dropped | Drop % | Remaining | Dropped | Drop % |
| Raw Dataset | 1,147 | 0 | 0.0 | 220 | 0 | 0.0 |
| Gene/Disease Mapping | 986 | 161 | 14.0 | 142 | 78 | 35.5 |
| Patient Creation | 986 | 0 | 0.0 | 142 | 0 | 0.0 |
| Multi-Gene Filtering | 981 | 5 | 0.5 | 114 | 28 | 19.7 |
| Deduplication | 960 | 21 | 2.1 | 114 | 0 | 0.0 |
| Quality Filtering | 930 | 30 | 3.1 | 114 | 0 | 0.0 |
| **Final Dataset** | **930** | **217** | **18.9** | **114** | **106** | **48.2** |

### A.2.3 SUBGRAPHS CREATION AND FILTERING

For each patient, we construct the candidate gene set by taking all genes that are within 2 hops of any of the patient's phenotypes in PrimeKG , and we use the union of these 2-hop neighborhoods as the search space. We use 2 hops as a practical default: 3-hop neighborhoods in PrimeKG quickly explode to 16–20k genes and make shortest-path construction and training computationally infeasible for our setup, while most of causal genes in our benchmarks already fall within 2 hops. To avoid artificially favoring our method, we exclude patients whose causal gene is not reachable within 2 hops, instead of injecting the true gene by hand. On top of this, we optionally apply a filtering step that keeps only genes appearing in at least 25% of phenotype-specific subgraphs for that patient, which helps remove spurious links from noisy phenotype annotations. This filtering is purely a hyperparameter, can be turned on or off, and we provide ablations (0% vs 10% vs 25%) showing that it slightly stabilizes performance but does not change the overall conclusions. Importantly, the 2-hop rule and filtering (when used) are applied identically across all datasets and baselines, preserving fairness of the comparison. Clinically, this setup is realistic and flexible. When clinicians have panel- or variant-derived gene lists, Tab. 2 reflects the expected performance. When no prior information is available, PhenoKG uses the 2-hop candidate set, which can be optionally expanded by the clinician. Thus, 2 hops is a default choice, not a hard limitation.

Table 7: Candidate set statistics and preprocessing summary across filtering levels. Raw patients and unreachable-in-2-hop counts are dataset-level constants; filtering levels vary in the number of candidates and usable patients.

| Dataset | Raw Patients | Not Reachable (2 hops) | Filtering (%) | Mean # Candidates | # Patients |
|---|---|---|---|---|---|
| PhenoDis | 112 | 1 | 0 | 6674.74 | 111 |
| | | | 10 | 3360.25 | 110 |
| | | | 25 | 1070.99 | 109 |
| MyGene2 | 146 | 10 | 0 | 3594.48 | 136 |
| | | | 10 | 2950.60 | 136 |
| | | | 25 | 1829.92 | 135 |
| FGDD | 930 | 157 | 0 | 1935.32 | 773 |
| | | | 10 | 1882.98 | 773 |
| | | | 25 | 1056.81 | 738 |
| Simulated | 320 | 3 | 0 | 6455.37 | 317 |
| | | | 10 | 3605.09 | 315 |
| | | | 25 | 634.46 | 306 |

## A.3 DATASET OVERLAP ANALYSIS

This section describes how the entity-level overlap, and patient-level similarity analyses were conducted and how the results should be interpreted in the context of data leakage and model generalization.

Table 8: Entity-level overlap between training and test sets. Coverage = fraction of test entities seen during training. Jaccard = |Intersection| / |Union|.

| Dataset | Patients | Phenotype Cov. | Gene Cov. | Phenotype Jaccard | Gene Jaccard |
|---------|----------|----------------|-----------|-------------------|--------------|
| PhenoDis | 109 | 0.885 | 0.721 | 0.186 | 0.028 |
| Simulated | 306 | 0.969 | 0.505 | 0.329 | 0.045 |
| Simulated (with cand. genes) | 320 | 0.970 | 0.505 | 0.336 | 0.046 |
| FGDD | 738 | 0.848 | 0.653 | 0.050 | 0.062 |
| MyGene2 | 135 | 0.941 | 0.786 | 0.044 | 0.015 |

### A.3.1 ENTITY-LEVEL OVERLAP COMPUTATION

Table 8 summarizes the overlap between the training dataset and each test dataset at the *entity* level of phenotypes (HPO terms) and genes. For each dataset, we extracted the set of phenotypes and genes present in the training data, and the corresponding sets in the test data.

**Coverage Metrics.** Coverage is defined as the proportion of test-set entities that were observed in the training data:

$$\text{Phenotype Coverage} = \frac{|P_{\text{train}} \cap P_{\text{test}}|}{|P_{\text{test}}|}, \qquad \text{Gene Coverage} = \frac{|G_{\text{train}} \cap G_{\text{test}}|}{|G_{\text{test}}|}.$$

These asymmetric metrics answer: What fraction of test phenotypes/genes were already seen during training?

**Jaccard Similarity.** To quantify the global similarity between training and test entity distributions, we used the Jaccard index:

$$J(P_{\text{train}}, P_{\text{test}}) = \frac{|P_{\text{train}} \cap P_{\text{test}}|}{|P_{\text{train}} \cup P_{\text{test}}|}, \qquad J(G_{\text{train}}, G_{\text{test}}) = \frac{|G_{\text{train}} \cap G_{\text{test}}|}{|G_{\text{train}} \cup G_{\text{test}}|}.$$

Unlike coverage, the Jaccard index is symmetric and accounts for the size of the combined entity space. Because the union sets are large, particularly for phenotypes, Jaccard values remain relatively low (0.18 to 0.33 for phenotypes and 0.015 to 0.062 for genes), even when coverage is high. This demonstrates that entity vocabularies overlap partially but do not coincide.

These results indicate that test datasets share some phenotype vocabulary with the training data, which is a common occurrence in rare-disease cohorts, but the global similarity between entity distributions is modest. Gene overlap is even lower, and no dataset exhibits substantial reuse of gene sets across training and test partitions.

Table 9: Patient profile similarity between training and test sets. All metrics are based on Jaccard similarity of phenotype sets.

| Dataset | Profiles | Mean Jaccard | Std | Max | 95th %ile |
|---------|----------|--------------|-----|-----|-----------|
| PhenoDis | 109 | 0.0068 | 0.0144 | 0.1600 | 0.0375 |
| Simulated | 306 | 0.0134 | 0.0211 | 0.2778 | 0.0556 |
| Simulated (with cand. genes) | 320 | 0.0125 | 0.0209 | 0.2703 | 0.0556 |
| FGDD | 738 | 0.0040 | 0.0134 | 0.2857 | 0.0333 |
| MyGene2 | 135 | 0.0075 | 0.0176 | 0.1579 | 0.0465 |

### A.3.2 PATIENT-PROFILE SIMILARITY COMPUTATION

While entity overlap is informative, the most important leakage risk arises when an entire patient profile in the test set closely resembles a profile in the training set. Table 9 reports such similarity using the Jaccard index over phenotype sets.

For each training patient $A_i$ and each test patient $B_j$, we compute:

$$J(A_i, B_j) = \frac{|A_i \cap B_j|}{|A_i \cup B_j|}.$$

We report the following metrics. Mean Jaccard and Standard Deviation: average similarity between training and test profiles; Maximum Similarity - the most similar training-test pair; 95th Percentile is a similarity threshold below which 95% of pairs fall.High-Similarity Percentage - proportion of pairs with $J > 0.7$.

Across all datasets, mean similarity values are extremely low ($< 0.014$), maximum similarities remain modest ($< 0.29$), and no training-test pair exhibits high similarity ($J > 0.7$) or very high similarity ($J > 0.9$). These results show that phenotype profiles differ substantially across datasets and that no near-duplicate patient records exist across training and test splits. The combined entity-level and patient-level analyses demonstrate that the model cannot benefit from memorization or overlap between training and test data. Although some degree of phenotype vocabulary overlap is expected in medical datasets, Jaccard values reveal that global similarity between training and test entity spaces is low. More importantly, patient-level similarity is effectively zero: no test patient shares a highly similar phenotype profile with any training patient. Thus, the evaluation reflects generalization to novel combinations of phenotypes and gene–phenotype relationships.

## A.4 ABLATION EXPERIMENTS ON FILTERING

To show that filtering is an optional hyperparameter to reduce noise in large HPO profiles, we conducted ablation experiments on FGDD and PhenoDis with 25%, 10% and without filtering , and the overall conclusions remain unchanged. We selected FGDD because, although the absolute number removed is still small, this dataset shows the highest proportion of filtered patients compared with the others and PhenoDis has the largest drop in candidate gene lists after filtering among the real-world datasets (Table 7). We did the comparsion only with PhenoApt, since according to the main results, this method showed the best performance among other competitive methods. Table 10 shows that for FGDD datset difference between 10% and 0% filtering is minimal and the performance really close with PhenoApt, exceeding it from Hits@10. For PhenoDis we created two Tables **??**, since the number of patients in each filtering is different. In both cases our method outperformed the PhenoApt on the PhenoDis dataset with the big margin. Overall, the filtering helps improve performance, as it removes noisy candidate genes that are linked to only a small fraction of phenotypes. The filtering step is optional and can be entirely removed if the clinician assigns equal importance to all phenotype terms and values them equally.

Table 10: Performance comparison between PhenoApt and our method on FGDD dataset across MRR and Hits@K metrics with different types of filtering. Values are shown as percentages.

| Model | MRR | Hits@1 | Hits@3 | Hits@5 | Hits@10 | Hits@15 | Hits@20 | Hits@25 | Hits@30 | Hits@50 | Hits@75 | Hits@100 |
|---|---|---|---|---|---|---|---|---|---|---|---|---|
| PhenoApt | **25.28** | **16.7** | **28.9** | **33.1** | 39.0 | 42.7 | 46.9 | 49.3 | 52.0 | 56.9 | 63.6 | 68.7 |
| Ours (10% filtering) | $24.4 \pm 0.5$ | $16.2 \pm 0.9$ | $26.5 \pm 0.4$ | $31.7 \pm 0.7$ | $41.4 \pm 0.1$ | $\mathbf{46.1 \pm 0.6}$ | $48.8 \pm 0.1$ | $51.2 \pm 0.5$ | $54.7 \pm 0.7$ | $\mathbf{62.0 \pm 0.5}$ | $66.5 \pm 0.3$ | $\mathbf{71.5 \pm 0.2}$ |
| Ours (0% filtering) | $24.5 \pm 0.6$ | $16.4 \pm 1.1$ | $26.3 \pm 0.4$ | $31.5 \pm 0.2$ | $\mathbf{41.4 \pm 0.3}$ | $46.1 \pm 0.5$ | $\mathbf{48.9 \pm 0.1}$ | $\mathbf{51.3 \pm 0.5}$ | $\mathbf{55.0 \pm 0.8}$ | $61.9 \pm 0.5$ | $\mathbf{66.6 \pm 0.3}$ | $71.5 \pm 0.1$ |

Table 11: Performance on the PhenoDis dataset (0% filtering). Values are shown as percentages.

| Model | MRR | Hits@1 | Hits@3 | Hits@5 | Hits@10 | Hits@15 | Hits@20 | Hits@25 | Hits@30 | Hits@50 | Hits@75 | Hits@100 |
|---|---|---|---|---|---|---|---|---|---|---|---|---|
| PhenoApt | 43.11 | 29.7 | 48.6 | 62.2 | 72.1 | 76.6 | 77.5 | 78.4 | 81.1 | 86.5 | 89.2 | 91.0 |
| Ours | $\mathbf{49.5 \pm 1.8}$ | $\mathbf{36.3 \pm 3.2}$ | $\mathbf{54.1 \pm 0.0}$ | $\mathbf{62.8 \pm 1.9}$ | $\mathbf{79.0 \pm 1.0}$ | $\mathbf{86.2 \pm 1.4}$ | $\mathbf{90.1 \pm 0.9}$ | $\mathbf{92.2 \pm 0.5}$ | $\mathbf{92.8 \pm 0.0}$ | $\mathbf{92.8 \pm 0.0}$ | $\mathbf{93.7 \pm 0.0}$ | $\mathbf{95.2 \pm 0.5}$ |

Table 12: Performance on the PhenoDis dataset (10% filtering). Values are shown as percentages.

| Model | MRR | Hits@1 | Hits@3 | Hits@5 | Hits@10 | Hits@15 | Hits@20 | Hits@25 | Hits@30 | Hits@50 | Hits@75 | Hits@100 |
|---|---|---|---|---|---|---|---|---|---|---|---|---|
| PhenoApt | 43.49 | 30.0 | 49.1 | 62.7 | 72.7 | 77.3 | 78.2 | 79.1 | 81.8 | 87.3 | 90.0 | 91.8 |
| Ours | $\mathbf{51.2 \pm 1.1}$ | $\mathbf{37.9 \pm 2.3}$ | $\mathbf{56.7 \pm 1.0}$ | $\mathbf{65.5 \pm 1.6}$ | $\mathbf{80.0 \pm 0.0}$ | $\mathbf{87.6 \pm 1.4}$ | $\mathbf{92.1 \pm 0.5}$ | $\mathbf{93.0 \pm 0.5}$ | $\mathbf{93.6 \pm 0.0}$ | $\mathbf{93.9 \pm 0.5}$ | $\mathbf{94.5 \pm 0.0}$ | $\mathbf{95.8 \pm 0.5}$ |

## A.5 WEIGHTING PARAMETERS $\hat{\alpha}, \hat{\beta}$

We set $\hat{\alpha} = 0.7$ and $\hat{\beta} = 0.3$ in the main experiments, based on validation set. We wanted to emphasize the new knowledge from the database, so we shifted the weighting toward a higher $\hat{\alpha}$. The

grid search in Table 5 shows the influence of these parameeters on the results, showing that balanced or phenotype-favored weights consistently outperform extreme cases.

### A.6 GPT-4O BASELINE SETUP

To establish a large language model (LLM) baseline, we evaluated GPT-4o (Achiam et al., 2023) via the OpenAI API. For each patient, we provided the list of positive phenotypes and all candidate genes as input, and prompted GPT-4o to return a ranked ordering of the candidate genes in JSON format. The exact prompt was:

> Patient ID: {patient_id}
> Phenotypes: {phenotype_list}
> Candidate genes: {candidate_list}
>
> Please rank the candidate genes from most to least likely to be causative. Return **only** a valid JSON object in the following format (no explanations, no markdown):
> { "id": "{patient_id}", "ranked_genes": ["GENE1", "GENE2", ...] }

The procedure was repeated for all patients 3 times in the test sets, and results were aggregated for evaluation.

### A.7 EVALUATION METRICS

We evaluated our model using Mean Reciprocal Rank (MRR) and Hits@$j$. MRR measures the inverse rank of the first correct gene for each patient and averages across all patients, providing a smooth sensitivity to ranking changes:

$$\text{MRR} = \frac{1}{|Q|} \sum_{q \in Q} \frac{1}{\text{rank}_q},$$

where $|Q|$ is the number of patients, and $\text{rank}_q$ is the position of the true causal gene in the ranked list for patient $q$.

Hits@$j$ measures the proportion of patients for which the correct gene appears among the top $j$ ranked genes:

$$\text{Hits@}j = \frac{1}{|Q|} \sum_{q \in Q} \mathbf{1}\{\text{rank}_q \leq j\},$$

where $\mathbf{1}\{\cdot\}$ is the indicator function.

While MRR reflects ranking quality across all cases, Hits@$j$ captures the success rate within the top-$j$ predictions, which is particularly relevant in practical prioritization scenarios.

## B USE OF LARGE LANGUAGE MODELS

We used an LLM solely to correct grammar and rephrase sentences for improved readability. All outputs were manually reviewed.

