# OpenReview forum: "PhenoKG: Knowledge Graph-Driven Gene Prioritization and Patient Insights from Phenotypes Alone"
_ICLR.cc/2026/Conference — Submitted to ICLR 2026_

### Official Review · Reviewer_9P6M · 2025-10-30

**Soundness:** 3
**Presentation:** 3
**Contribution:** 2
**Rating:** 6
**Confidence:** 3

**Summary:**

The paper tackles phenotype-driven rare-disease gene prioritization by building patient-specific subgraphs over a biomedical knowledge graph and encoding them with GATv2 plus transformer modules, followed by a contrastive ranking objective and an optional phenotype-based reranking step. Across simulated and real-world cohorts, the method reports strong MRR/Hits@K, outperforming several phenotype/KG baselines in both candidate-list and no-candidate settings. Overall the problem is important and the empirical results are promising.

**Strengths:**

- The proposed representation connects biomedical KGs with clinical phenotypes in a flexible, modular way.

- The approach is evaluated on multiple datasets (three real-world plus a simulated cohort) and two usage scenarios (with/without candidate lists), which improves the breadth and credibility of the results.

**Weaknesses:**

- Because the candidate list is predefined in many clinical settings, it would be helpful to discuss whether the framework can discover new, previously unlisted genes for a patient/phenotype (beyond reranking within a fixed set).
- The data preprocessing is fairly involved; this may filter out difficult edge cases and could limit realism. A concise flowchart with counts at each step would make exclusions transparent.
- Please add a table of basic dataset statistics (before/after preprocessing), e.g., average number of phenotypes per patient and the distribution of candidate list sizes.

**Questions:**

- Please report potential overlap between training and test distributions at both the entity level (genes/phenotypes appearing in both) and the association/patient-profile level (e.g., patients with very similar phenotype sets), and discuss how this overlap might affect generalization.

---

> ### Author Response · Authors · 2025-11-21
> **Rebuttal by Authors**
>
> We thank reviewer for the constructive and thoughtful evaluation. We appreciate the reviewer’s recognition of the importance of the problem, the strength of our empirical results across multiple datasets, and the flexibility of our method. The reviewer also raised several valid suggestions regarding preprocessing transparency and dataset statistics. We will incorporate all requested preprocessing details, add summary tables, and include a clear flowchart with counts at each step in the revised manuscript.
>
> **W1. Novel-gene Discovery.** Theoretically, PhenoKG should be able to surface previously unlisted genes, and this is one of the reasons we designed the framework to support genome-wide prioritization rather than relying solely on predefined clinical lists. However, making a strong claim about true novel-gene discovery requires dedicated evaluation with clinicians and controlled experiments (e.g., masking known edges or perturbing the KG at least). This is planned for our follow-up work.
>
> **W2. Preprocessing Steps and Potential Filtering Biases.**
>
> We thank the reviewer for the comment. We will add a flowchart-style table in the supplement reporting sample counts at each preprocessing step for full transparency. We agree only partially with the concern. Some preprocessing is necessary, but it does not selectively remove “difficult” cases since it follows directly from the structure of the KG and how phenotypes and genes are connected. The 2-hop gene set is used for computational feasibility; the framework can operate with any candidate list, and clinicians may extend it with additional genes or panels as needed.
>
> To ensure a fair evaluation for all methods, we include all genes reachable within 2 hops and exclude only cases where the true gene is not reachable in this space. Otherwise, one would need to artificially insert the true gene, which is common in candidate-gene–based evaluations but is not a principled strategy. Our rule is applied uniformly across all baselines.
> Clinically, this setup is realistic and flexible:
> - When clinicians have panel- or variant-derived gene lists, Tab. 2 reflects the expected performance.
> - When no prior information is available, PhenoKG uses the 2-hop candidate set, which can be optionally expanded by the clinician. Thus, 2 hops is a default choice, not a hard limitation.
>
> **Further Clarifications**:
> 3-hop neighborhoods would expand to ∼16-20k genes due to the density of PrimeKG and make shortest-path construction computationally infeasible for us. Methodologically, the model can use 3 hops; this is purely a resource constraint.
> Filtering is an optional hyperparameter to reduce noise in large HPO profiles. We include ablation table without filtering in the supplement, and the overall conclusions remain unchanged.  We selected FGDD because, although the absolute number removed is still small, this dataset shows the highest proportion of filtered patients compared with the others.
>
> | Model               | MRR          | Hits@1       | Hits@3       | Hits@5       | Hits@10      | Hits@15      | Hits@20      | Hits@25      | Hits@30      | Hits@50      | Hits@75      | Hits@100     |
> |---------------------|--------------|--------------|--------------|--------------|--------------|--------------|--------------|--------------|--------------|--------------|--------------|--------------|
> | PhenoApt            | 25.28        | 16.7         | 28.9         | 33.1         | 39.0         | 42.7         | 46.9         | 49.3         | 52.0         | 56.9         | 63.6         | 68.7         |
> | Ours 10% filtering  | 24.4 ± 0.5   | 16.2 ± 0.9   | 26.5 ± 0.4   | 31.7 ± 0.7   | 41.4 ± 0.1   | 46.1 ± 0.6   | 48.8 ± 0.1   | 51.2 ± 0.5   | 54.7 ± 0.7   | 62.0 ± 0.5   | 66.5 ± 0.3   | 71.5 ± 0.2   |
> | Ours 0% filtering   | 24.5 ± 0.6   | 16.4 ± 1.1   | 26.3 ± 0.4   | 31.5 ± 0.2   | 41.4 ± 0.3   | 46.1 ± 0.5   | 48.9 ± 0.1   | 51.3 ± 0.5   | 55.0 ± 0.8   | 61.9 ± 0.5   | 66.6 ± 0.3   | 71.5 ± 0.1   |
>
>
> Finally, all methods were evaluated on the same datasets with identical preprocessing, ensuring fairness and rigor in the comparison.

---

> ### Author Response · Authors · 2025-11-21
>
> **W3. Dataset Statistics.** We thank the reviewer for the valid comment. We will add the following tables to the manuscript as well.
>
> | Dataset    | Raw Patients | Not Reachable (2 hops) | Filtering (%) | Mean # Candidates Genes | # Patients |
> |------------|--------------|-------------------------|----------------|--------------------|------------|
> | **PhenoDis** | **112**        | **1**                     | 0              | 6674.74            | 111        |
> |            |              |                         | 10             | 3360.25            | 110        |
> |            |              |                         | 25             | 1070.99            | 109        |
> | **MyGene2**  | **146**        | **10**                    | 0              | 3594.48            | 136        |
> |            |              |                         | 10             | 2950.60            | 136        |
> |            |              |                         | 25             | 1829.92            | 135        |
> | **FGDD**     | **930**        | **157**                   | 0              | 1935.32            | 773        |
> |            |              |                         | 10             | 1882.98            | 773        |
> |            |              |                         | 25             | 1056.81            | 738        |
> | **Simulated**| **320**        | **3**                     | 0              | 6455.37            | 317        |
> |            |              |                         | 10             | 3605.09            | 315        |
> |            |              |                         | 25             | 634.46             | 306        |
>
>
> | Dataset                  | # Patients | # Phenotypes / Patient | # Unique Phenotypes | # Unique Genes |
> |--------------------------|----------|------------------------|--------------------|--------------|
> | Training Dataset         | 36,224   | 18.5 ± 7.7             | 6,152              | 2,164        |
> | FGDD                     | 738      | 7.3 ± 4.0              | 369                | 213          |
> | Simulated (full)         | 320      | 17.8 ± 7.0             | 2,152              | 208          |
> | Simulated (2-hop valid)  | 306      | 18.0 ± 7.0             | 2,109              | 200          |
> | MyGene2                  | 135      | 8.3 ± 6.7              | 288                | 42           |
> | PhenoDis                 | 109      | 30.4 ± 36.3            | 1,327              | 86           |
> | **All Test Sets**        | 1,288    | -                      | 3,021              | 514          |

---

> > ### Author Response · Authors · 2025-11-21
> >
> > **Q1. Train–test Overlap and Leakage Analysis**. We thank the reviewer for raising this important point. We conducted a detailed overlap analysis at both the entity level and the patient-profile level, and the results confirm that the evaluation is not affected by hidden leakage.
> >
> > To compute these numbers, we extracted the phenotype and gene sets from the training split and each test dataset, computed coverage (fraction of test entities seen in training), and computed Jaccard similarity over the union of training and test entity vocabularies. For patient-level similarity, we compared each sampled test patient to each sampled training patient and computed the Jaccard index over phenotype sets, reporting mean, max, 95th percentile.
> >
> > At the entity level, test datasets share only partial vocabulary with the training data. While phenotype and gene coverage is moderate (e.g., 0.85–0.97 for phenotypes), the Jaccard similarity is low across all datasets (0.18–0.33 for phenotypes and 0.015–0.062 for genes;). This indicates that the global distribution of genes and phenotypes in the test sets differs substantially from the training set.
> >
> > | Dataset                              | Patients | Phenotype Cov. | Gene Cov. | Phenotype Jaccard | Gene Jaccard |
> > |--------------------------------------|----------|------------------|-----------|--------------------|--------------|
> > | PhenoDis                             | 109      | 0.885            | 0.721     | 0.186              | 0.028        |
> > | Simulated                            | 306      | 0.969            | 0.505     | 0.329              | 0.045        |
> > | Simulated (with cand. genes)         | 320      | 0.970            | 0.505     | 0.336              | 0.046        |
> > | FGDD                                 | 738      | 0.848            | 0.653     | 0.050              | 0.062        |
> > | MyGene2                              | 135      | 0.941            | 0.786     | 0.044              | 0.015        |
> >
> > More importantly, at the patient-profile level, we observe near-zero similarity between any training–test pair. Mean Jaccard similarities are <0.014, maximum similarity is <0.29, and no pair exceeds 0.7. This shows that no test patient has a near-duplicate or highly similar phenotype profile in the training data.
> > Together, these analyses demonstrate that PhenoKG is evaluated on novel combinations of phenotypes and genes, and that performance cannot be attributed to memorization or unintended overlap. We will include the condensed version of this analysis (with tables) in the final manuscript.
> >
> > | Dataset                              | Profiles | Mean Jaccard | Std     | Max     | 95th %ile |
> > |--------------------------------------|----------|--------------|---------|---------|-----------|
> > | PhenoDis                             | 109      | 0.0068       | 0.0144  | 0.1600  | 0.0375    |
> > | Simulated                            | 306      | 0.0134       | 0.0211  | 0.2778  | 0.0556    |
> > | Simulated (with cand. genes)         | 320      | 0.0125       | 0.0209  | 0.2703  | 0.0556    |
> > | FGDD                                 | 738      | 0.0040       | 0.0134  | 0.2857  | 0.0333    |
> > | MyGene2                              | 135      | 0.0075       | 0.0176  | 0.1579  | 0.0465    |

---

> ### Author Response · Authors · 2025-11-26
> **Additional experiments to address the potential filtering biases.**
>
> **W2. Preprocessing Steps and Potential Filtering Biases.**
>
> We also extended the evaluation to the PhenoDis dataset, as it shows the largest drop in candidate gene lists after filtering among the real-world datasets. We compare only with PhenoApt, since the main results show that this method achieves the strongest performance among the competitive baselines.
> For PhenoDis, we present two separate tables, because the number of evaluated patients differs across filtering settings. In both cases, our method outperforms PhenoApt by a large margin.
> Filtering improves performance by removing noisy candidate genes linked to only a small fraction of phenotypes. This step is optional and can be omitted if the clinician considers all phenotype terms equally important.
>
> | **Model** | **MRR** | **Hits@1** | **Hits@3** | **Hits@5** | **Hits@10** | **Hits@15** | **Hits@20** | **Hits@25** | **Hits@30** | **Hits@50** | **Hits@75** | **Hits@100** |
> |----------|--------:|-----------:|-----------:|-----------:|------------:|------------:|------------:|------------:|------------:|-----------:|-----------:|------------:|
> | PhenoApt | 43.11   | 29.7       | 48.6       | 62.2       | 72.1        | 76.6        | 77.5        | 78.4        | 81.1        | 86.5       | 89.2       | 91.0        |
> | Ours     | 49.5 ± 1.8 | 36.3 ± 3.2 | 54.1 ± 0.0 | 62.8 ± 1.9 | 79.0 ± 1.0 | 86.2 ± 1.4 | 90.1 ± 0.9 | 92.2 ± 0.5 | 92.8 ± 0.0 | 92.8 ± 0.0 | 93.7 ± 0.0 | 95.2 ± 0.5 |
>
>
> | **Model** | **MRR** | **Hits@1** | **Hits@3** | **Hits@5** | **Hits@10** | **Hits@15** | **Hits@20** | **Hits@25** | **Hits@30** | **Hits@50** | **Hits@75** | **Hits@100** |
> |----------|--------:|-----------:|-----------:|-----------:|------------:|------------:|------------:|------------:|------------:|-----------:|-----------:|------------:|
> | PhenoApt | 43.49   | 30.0       | 49.1       | 62.7       | 72.7        | 77.3        | 78.2        | 79.1        | 81.8        | 87.3       | 90.0       | 91.8        |
> | Ours     | 51.2 ± 1.1 | 37.9 ± 2.3 | 56.7 ± 1.0 | 65.5 ± 1.6 | 80.0 ± 0.0 | 87.6 ± 1.4 | 92.1 ± 0.5 | 93.0 ± 0.5 | 93.6 ± 0.0 | 93.9 ± 0.5 | 94.5 ± 0.0 | 95.8 ± 0.5 |

---

> > ### Comment · Reviewer_9P6M · 2025-11-26
> > **Response to the rebuttal**
> >
> > Thanks for the thorough quantitative analyses. The rebuttal satisfactorily addresses my concerns; I have no further comments at this time and will raise my score accordingly.

---

### Official Review · Reviewer_Fp52 · 2025-10-31

**Soundness:** 2
**Presentation:** 3
**Contribution:** 2
**Rating:** 4
**Confidence:** 3

**Summary:**

This paper tackles the challenge of rare genetic disease diagnosis using patient phenotypes alone, aiming to infer disease-causing genes through the relationships among phenotypes, diseases, and genes. The authors propose PhenoKG, a knowledge-graph–based framework that integrates graph neural networks and transformer encoders to model phenotype–gene associations. It constructs a patient-specific subgraph from phenotype–gene associations within a biomedical knowledge graph (PrimeKG) and learns patient and gene embeddings for ranking ∼6,000 genes by their likelihood of being causative. The method also includes a reranking step using external databases (e.g., Monarch) to incorporate up-to-date phenotype–gene associations. Experiments on multiple datasets (FGDD, PhenoDis, MyGene2, and simulated cohorts) show that PhenoKG achieves state-of-the-art performance both with and without candidate gene lists, outperforming methods such as Shepherd, PhenoApt, and Amelie.

**Strengths:**

A key strength of the paper lies in its integration of graph neural networks with transformer-based encoders, which allows the model to effectively capture rich, patient-specific features as well as characteristics of potential causal genes. By leveraging these complementary architectures, PhenoKG is able to align patient and gene representations across modalities, enabling more precise modeling of phenotype–gene relationships and improving the prioritization of candidate causal genes.

**Weaknesses:**

One concern is the scalability of the approach. As the authors note, there are thousands of potential causal genes in this task, and computing embeddings for all genes could be computationally intensive. This may limit the practicality of PhenoKG in real-world scenarios where large gene sets need to be evaluated.

Another concern relates to the model’s performance consistency across datasets. While PhenoKG achieves state-of-the-art results on most benchmarks, it performs substantially worse than PhenoApt on the MyGene2 dataset. Although the authors provide some analysis of the dataset’s characteristics and potential reasons for this discrepancy, it raises questions about the generalizability of the approach and suggests that further investigation into dataset-specific performance is warranted.

**Questions:**

1. Candidate Gene Extraction: The paper briefly states that candidate genes are extracted “based on phenotype–gene associations,” but the specific algorithmic procedure (e.g., thresholds, ontology traversal strategy) is not detailed. How exactly are the candidate gene lists extracted when no clinician-provided list is available? Is there a fixed k-hop depth or a phenotype–gene similarity threshold?

2. Could the authors provide component-level ablations—for example, evaluating the effectiveness of the reranking module, and whether integrating external knowledge bases (e.g., Monarch) actually improves results?

3. Can the authors provide an example of a real patient case study showing how PhenoKG identified the correct gene and what features contributed most?

4. The authors mention that they imposed multiple restrictions for efficiency, such as limiting the knowledge graph to a 2-hop neighborhood around each patient and only considering genes that appear in at least 10% of the patient’s phenotype subgraphs as candidates. More details are needed to understand how these restrictions might affect the fairness and rigor of the evaluation—for example, clarifying what exactly “2-hop” refers to and how it is defined in patient knowledge graph. A clear explanation of these choices would strengthen confidence in the fairness and validity of the reported results.

5.  Could the authors show the results over some widely used datasets, such as DDD and the datasets used in LIRICAL, Phen2Gene and PhenoApt?  It can illustrate the robustness of the proposed method.

---

> ### Author Response · Authors · 2025-11-21
> **Rebuttal by Authors**
>
> We thank the reviewer for the detailed and insightful review. We appreciate the strengths noted regarding the integration of graph neural networks and transformer-based encoders. We address the main raised points here:
>
> **W1. Scalability.**
> - PhenoKG is designed to handle thousands of candidate genes. We use 2-hop candidate extraction because 3-hop neighborhoods would expand to roughly 16–20k genes, and computing shortest paths on subgraphs of that size was not computationally feasible for us. This is a resource limitation, not a limitation of the method—the framework can operate with 3 hops if resources permit.
> - Clinically, starting from all ~20k genes is uncommon. Clinicians usually rely on HPO-based panels or prior suspicion, and 2 hops already captures the relevant search space in almost all cases. Moreover, if a clinician wants certain genes included, any gene or full panel can be added directly. The 2-hop set is therefore a practical default, not a restriction of PhenoKG.
>
> **W2. Cross-data Performance**. We agree that the lower performance on MyGene2 is a valid concern. MyGene2 has very sparse phenotype annotations and is strongly biased toward highly reported genes, which benefits frequency-based tools. To address this, we updated the MyGene2 results by incorporating external phenotype–gene associations from the Monarch API and the HPO annotation dataset [b]  into the post-filtering step. After adding this information, the performance on MyGene2 improved substantially, and the updated numbers are included in the revision (Tab. 4).
>
> | **Filtering Method (MyGene2)** | **MRR** | **Hits@1** | **Hits@3** | **Hits@5** | **Hits@10** | **Hits@15** | **Hits@20** | **Hits@25** | **Hits@30** | **Hits@50** | **Hits@75** | **Hits@100** |
> |-------------------------------|---------|------------|------------|------------|-------------|-------------|-------------|-------------|-------------|-------------|-------------|--------------|
> | Ours — Monarch only | 31.1 ± 1.8 | 22.5 ± 1.1 | 32.3 ± 3.5 | 37.8 ± 3.4 | 51.4 ± 3.5 | 56.0 ± 1.1 | 64.2 ± 6.8 | 69.4 ± 2.4 | 73.1 ± 2.6 | 79.0 ± 1.1 | 81.9 ± 1.5 | 85.4 ± 0.9 |
> | Ours — Monarch + HPO associations | **50.9 ± 3.7** | **40.7 ± 5.2** | **55.6 ± 3.4** | **63.2 ± 5.2** | **74.8 ± 0.7** | **77.3 ± 0.9** | **78.3 ± 1.1** | **79.0 ± 0.4** | **79.8 ± 0.4** | **84.2 ± 0.9** | **87.7 ± 1.1** | **89.6 ± 0.0** |
>
>
> This indicates that the limitation arises primarily from incomplete KG coverage for this specific dataset, and extending the KG with additional sources mitigates the issue. We will clarify this point in the manuscript.
>
> [b] Human Phenotype Ontology. HPO Annotations Dataset.
> Available at: https://hpo.jax.org/data/annotations

---

> > ### Author Response · Authors · 2025-11-21
> >
> > **Combined Response to Q1 and Q4 (Candidate Gene Extraction Procedure Design Choice Clarifications)**
> > When no clinician-provided list is available, we extract candidate genes by taking all genes within 2 hops of any patient phenotype in PrimeKG (Sec. 3.2). Candidate genes are therefore the union of these 2-hop neighborhoods across the patient’s phenotypeWe use k = 2 because 3-hop subgraphs would expand the gene set to roughly 16–20k genes (not counting other node types), which is not computationally feasible. In practice, based on the training datasets, almost all causal genes already fall within 2 hops of their phenotypes. This restriction is applied uniformly across all datasets and baselines, and we exclude patients whose causal gene does not appear within 2 hops, as noted in the paper. Using 3 hops does not change the method conceptually. No phenotype–gene similarity threshold is applied.
> >
> > We optionally apply a filtering step that keeps only genes appearing in at least 25% (10% was a mistake in the paper) of the patient’s phenotype-specific subgraphs; this stabilizes phenotype–gene associations by removing spurious links. The filtering step is optional and was introduced only to mitigate noise in phenotype annotations: a gene is kept if it appears in at least 25% of the phenotype-specific subgraphs (Sec. 4.1). This step is purely a hyperparameter and can be omitted; the system can work without filtering as well. We included an ablation table showing how performance changes with and without filtering on the FGDD dataset using the same 773 patients (for 25% it was 738 patients). We selected FGDD because, although the absolute number removed is still small, this dataset shows the highest proportion of filtered patients compared with the others.
> >
> > | Model               | MRR          | Hits@1       | Hits@3       | Hits@5       | Hits@10      | Hits@15      | Hits@20      | Hits@25      | Hits@30      | Hits@50      | Hits@75      | Hits@100     |
> > |---------------------|--------------|--------------|--------------|--------------|--------------|--------------|--------------|--------------|--------------|--------------|--------------|--------------|
> > | PhenoApt            | 25.28        | 16.7         | 28.9         | 33.1         | 39.0         | 42.7         | 46.9         | 49.3         | 52.0         | 56.9         | 63.6         | 68.7         |
> > | Ours 10% filtering  | 24.4 ± 0.5   | 16.2 ± 0.9   | 26.5 ± 0.4   | 31.7 ± 0.7   | 41.4 ± 0.1   | 46.1 ± 0.6   | 48.8 ± 0.1   | 51.2 ± 0.5   | 54.7 ± 0.7   | 62.0 ± 0.5   | 66.5 ± 0.3   | 71.5 ± 0.2   |
> > | Ours 0% filtering   | 24.5 ± 0.6   | 16.4 ± 1.1   | 26.3 ± 0.4   | 31.5 ± 0.2   | 41.4 ± 0.3   | 46.1 ± 0.5   | 48.9 ± 0.1   | 51.3 ± 0.5   | 55.0 ± 0.8   | 61.9 ± 0.5   | 66.6 ± 0.3   | 71.5 ± 0.1   |
> >
> >
> > From a clinical perspective, this filtering is a hyperparameter: if a clinician wants stronger alignment with the phenotype list, filtering can be enabled; if they suspect that the causal gene might be linked to only one phenotype, filtering can be disabled. Because the threshold is relative (percentage-based), patients with a small number of HPO terms typically do not lose any candidates.
> >
> > Fairness of the evaluation is preserved because all models operate on the same patient information after the 2 hops and filtering, and the filtering criteria (when used) are applied identically across methods and datasets.
> > We will make these points clearer in the revised manuscript.

---

> ### Author Response · Authors · 2025-11-21
>
> **Q2. Component-level ablations (including reranking)**. We did provide component-level ablations. The effect of the reranking step is reported in Tab. 4 (Appx. A). The effects of dual loss, embeddings, and graph construction are also ablated in Appx. A. We will explicitly cross-reference these results in Sec. 4 to make this clearer.
>
> **Q3. Real Patient Case Study.** Due to space constraints and patient privacy considerations, we did not include detailed case-level analyses in the conference version. We plan to include full case studies (with clinical narratives and model output explanation analysis) in an extended journal version.
>
> **Q5. Inclusion of DDD and other datasets.**
>
> We thank the reviewer for the suggestion to include results on additional widely used datasets such as DDD and those used in LIRICAL, Phen2Gene, and PhenoApt. We now include Phen2Gene as a baseline and have updated the tables accordingly.
>
> | Dataset      | Model       | MRR             | Hits@1         | Hits@3         | Hits@5         | Hits@10        | Hits@15        | Hits@20        | Hits@25        | Hits@30        | Hits@50        | Hits@75        | Hits@100       |
> |--------------|-------------|------------------|----------------|----------------|----------------|----------------|----------------|----------------|----------------|----------------|----------------|----------------|----------------|
> | **PhenoDis** | Phen2Gene   | 34.7             | 25.7           | 35.8           | 45.0           | 56.9           | 57.8           | 63.3           | 66.1           | 67.9           | 82.6           | 88.1           | 91.7           |
> |              | Ours        | 62.3 ± 3.3       | 50.8 ± 6.6     | 69.4 ± 1.4     | 79.8 ± 1.6     | 85.6 ± 0.5     | 86.2 ± 0.0     | 87.8 ± 1.1     | 89.0 ± 0.9     | 90.5 ± 0.5     | 92.7 ± 0.9     | 93.6 ± 0.9     | 94.2 ± 1.1     |
> | **FGDD**      | Phen2Gene   | 16.1             | 10.0           | 18.7           | 21.8           | 27.4           | 32.3           | 33.7           | 35.4           | 36.7           | 42.0           | 49.5           | 54.3           |
> |              | Ours        | 25.7 ± 1.0       | 17.0 ± 1.6     | 27.8 ± 0.1     | 34.3 ± 0.5     | 44.3 ± 0.6     | 48.6 ± 1.8     | 53.2 ± 0.2     | 55.5 ± 1.0     | 57.7 ± 0.5     | 64.8 ± 0.7     | 70.9 ± 0.3     | 75.1 ± 0.2     |
> | **MyGene2**   | Phen2Gene   | 12.5             | 9.6            | 11.9           | 14.8           | 17.8           | 20.0           | 21.5           | 25.2           | 26.7           | 40.0           | 47.4           | 49.6           |
> |              | Ours        | 50.9 ± 3.7       | 40.7 ± 5.2     | 55.6 ± 3.4     | 63.2 ± 5.2     | 74.8 ± 0.7     | 77.3 ± 0.9     | 78.3 ± 1.1     | 79.0 ± 0.4     | 79.8 ± 0.4     | 84.2 ± 0.9     | 87.7 ± 1.1     | 89.6 ± 0.0     |
> | **Simulated** | Phen2Gene   | 26.7             | 16.9           | 28.8           | 38.6           | 49.0           | 54.3           | 58.8           | 62.1           | 64.1           | 72.6           | 77.1           | 79.4           |
> |              | Ours        | 32.0 ± 1.9       | 22.5 ± 2.5     | 35.3 ± 1.3     | 42.6 ± 1.0     | 50.5 ± 1.8     | 54.8 ± 2.0     | 58.8 ± 0.9     | 61.1 ± 0.9     | 63.3 ± 1.2     | 70.8 ± 1.9     | 76.3 ± 0.2     | 80.6 ± 0.5     |
>
>
> We cannot include DDD or other controlled-access and in-house clinical cohorts because we do not have access to the required genomic data; DDD and the datasets used in Phen2Gene  rely on EGA-controlled variant files, filtering pipelines, and curated labels that are not publicly available. LIRICAL also cannot be used as a baseline because it performs disease-level, not gene-level, prioritization, and it does not output calibrated gene scores—mapping diseases back to genes would be invalid. Although Phen2Gene supports phenotype-only input, its original evaluation datasets are not reproducible for the same reasons. PhenoKG is designed for phenotype-only gene ranking, so we include evaluation of Phen2Gene with public datasets. All comparisons across reproducible methods (Amelie, Phen2Gene, PhenoApt, Shepherd) are fair, since we use the official implementations or APIs provided by the authors and evaluate every method on the same publicly available datasets.

---

> > ### Author Response · Authors · 2025-11-26
> > **Additional experiments to address the potential filtering biases.**
> >
> > **Combined Response to Q1 and Q4 (Candidate Gene Extraction Procedure Design Choice Clarifications)**
> >
> > We also extended the evaluation to the PhenoDis dataset, as it shows the largest drop in candidate gene lists after filtering among the real-world datasets. We compare only with PhenoApt, since the main results show that this method achieves the strongest performance among the competitive baselines.
> > For PhenoDis, we present two separate tables, because the number of evaluated patients differs across filtering settings. In both cases, our method outperforms PhenoApt by a large margin.
> > Filtering improves performance by removing noisy candidate genes linked to only a small fraction of phenotypes. This step is optional and can be omitted if the clinician considers all phenotype terms equally important.
> >
> > | **Model** | **MRR** | **Hits@1** | **Hits@3** | **Hits@5** | **Hits@10** | **Hits@15** | **Hits@20** | **Hits@25** | **Hits@30** | **Hits@50** | **Hits@75** | **Hits@100** |
> > |----------|--------:|-----------:|-----------:|-----------:|------------:|------------:|------------:|------------:|------------:|-----------:|-----------:|------------:|
> > | PhenoApt | 43.11   | 29.7       | 48.6       | 62.2       | 72.1        | 76.6        | 77.5        | 78.4        | 81.1        | 86.5       | 89.2       | 91.0        |
> > | Ours     | 49.5 ± 1.8 | 36.3 ± 3.2 | 54.1 ± 0.0 | 62.8 ± 1.9 | 79.0 ± 1.0 | 86.2 ± 1.4 | 90.1 ± 0.9 | 92.2 ± 0.5 | 92.8 ± 0.0 | 92.8 ± 0.0 | 93.7 ± 0.0 | 95.2 ± 0.5 |
> >
> >
> > | **Model** | **MRR** | **Hits@1** | **Hits@3** | **Hits@5** | **Hits@10** | **Hits@15** | **Hits@20** | **Hits@25** | **Hits@30** | **Hits@50** | **Hits@75** | **Hits@100** |
> > |----------|--------:|-----------:|-----------:|-----------:|------------:|------------:|------------:|------------:|------------:|-----------:|-----------:|------------:|
> > | PhenoApt | 43.49   | 30.0       | 49.1       | 62.7       | 72.7        | 77.3        | 78.2        | 79.1        | 81.8        | 87.3       | 90.0       | 91.8        |
> > | Ours     | 51.2 ± 1.1 | 37.9 ± 2.3 | 56.7 ± 1.0 | 65.5 ± 1.6 | 80.0 ± 0.0 | 87.6 ± 1.4 | 92.1 ± 0.5 | 93.0 ± 0.5 | 93.6 ± 0.0 | 93.9 ± 0.5 | 94.5 ± 0.0 | 95.8 ± 0.5 |

---

### Official Review · Reviewer_PmbE · 2025-10-31

**Soundness:** 3
**Presentation:** 3
**Contribution:** 2
**Rating:** 4
**Confidence:** 3

**Summary:**

The paper proposes PhenoKG, a knowledge graph (KG)–based framework for identifying causal genes given a patient’s phenotype and an external biomedical KG. It also introduces an optional reranking procedure that integrates recently validated clinical associations to expand the KG while maintaining robustness to noisy or incomplete input data. PhenoKG reportedly outperforms existing approaches across one simulated and three real rare-disease benchmarks. Ablation studies were performed to examine the effects of different loss functions, embedding initializations, and input graph configurations.


The paper addresses an important problem. However, I find that the methodological contributions are not clearly articulated, and the real-world impact of the proposed approach remains limited to benchmark demonstrations. A more thorough exposition and justification of the method’s design and novelty are required before publication. My detailed comments are as follows:



- Line 41: The distinction between disease-causal genes and genes associated with disease progression needs clarification. These two categories are known to have limited overlap (see “Limited overlap between genetic effects on disease susceptibility and disease survival”).

- Line 70: why 6,000 genes, not 20,000?

- Section 2: It is unclear how PhenoKG is methodologically distinct from prior work. If it represents a completely new framework, the authors need to justify its overall design choices. Furthermore, it is not clear which specific component drives the reported performance improvements.

- Lines 74–83: What is the main methodological innovation? If the reranking strategy constitutes the novelty, ablation experiments should explicitly demonstrate its contribution. This is related to line 422.

- Lines 308–323: The reranking procedure is presented as a key innovation, but the description does not establish why it is theoretically or methodologically principled.

- Line 294: Can the authors provide statistical significance measures (e.g., p-values) for the inferred causal genes for each patient?

**Strengths:**

See summary above.

**Weaknesses:**

See summary above.

**Questions:**

See summary above.

---

> ### Author Response · Authors · 2025-11-21
> **Rebuttal by Authors**
>
> We thank the reviewer for the thoughtful and detailed evaluation of our work. We appreciate the recognition of the importance of the problem and the helpful questions regarding our methodological clarity and design choices. We address the raised points below:
>
> **Q1, Q4. Methodological Contributions (Summary, Sec. 2)**. PhenoKG is inspired by Shepherd in using a KG as background knowledge and in learning patient representations, but it differs in several key aspects that define it as a new framework:
>
> **Patient-specific subgraph construction**: Shepherd operates on a global KG and requires a predefined candidate gene list. PhenoKG instead builds a new subgraph for each patient using shortest phenotype–gene paths from 2-hop genes (Sec. 3.2). This enables inference without candidate lists, which Shepherd cannot do. Even when we provide the same amount of genes for it, the performance of Shepherd is quite lower (Tab. 1).
>
> **Gene encoder redesign**: Shepherd embeds genes based on KG embeddings. PhenoKG introduces a Transformer-based gene encoder that processes gene embeddings jointly within each patient graph (Sec. 3.2.1). This architecture is new within phenotype-driven gene prioritization and is one of the key contributors to performance.
>
> **Dual contrastive objective with memory bank**: Shepherd trains only via link prediction and supervised ranking. PhenoKG adds a patient-similarity loss with a memory bank (Sec. 3.3), to encourage consistency across batches and improve generalization.  Ablations in Appx. A Tab. 3  and Figure 3 show the influence of each component.
>
> **Inference-time reranking**: Shepherd cannot incorporate new phenotype–gene associations at inference time. PhenoKG supports dynamic reranking using external evidence (Sec. 3.4), allowing updates without retraining. Ablations in Appx. A, Tab. 4 show clear gains from this component.
>
> **Flexibility in candidate gene settings**: Shepherd requires a curated candidate list and cannot operate over thousands of genes. PhenoKG works with large candidate sets ( Tab. 1) and small clinician-curated lists (Tab. 2).
>
>
> We will clarify these methodological distinctions more explicitly in the revised version.
>
>
> **Q2. Disease/Gene Causality and Association (L41)**. We thank the reviewer for the comment. We agree that disease-causal genes and genes influencing disease progression represent distinct categories with limited overlap, as also emphasized in the referenced study (“Limited overlap between genetic effects on disease susceptibility and disease survival,” Nat. Genet. 2025). Our work focuses exclusively on causal genes for monogenic disorders, not modifiers of disease course. This is reflected in datasets used in our work, each of which provides a single pathogenic gene per patient (Sec 4.1), and in our modeling objective, which treats only this gene as the positive target during training (Sec. 3.3). The sentence at Line 41 will be revised to clearly restrict our scope to identifying the gene responsible for initiating the disease, consistent with clinical practice in rare Mendelian diagnostics.
>
> **Q3. Number of Genes (L70)**. We thank the reviewers for raising this point. We will clarify this in the paper. The model is not limited to 4,000 genes (6k was a mistake). We include all gene nodes within 2 hops of the patient’s phenotypes in PrimeKG (Sec. 3.2), which results in roughly 4,000 genes depending on the dataset. Using 3 hops is possible in principle, but currently not computationally feasible, since 3 hops would be approximately 16-20k genes; this is a hyperparameter rather than a methodological constraint. The system is designed to be flexible: clinicians may add any additional candidate genes, including full panel-based lists or diagnostic guesses, and these are incorporated directly into the patient subgraph. PhenoKG therefore works both with large candidate set (thousands of genes; Tab. 1) and with small clinician-curated lists (Tab. 2), which, to our knowledge, is not supported by prior KG-based methods at the same time, it can be used as a prefiltering step and as a gene prioritization method.

---

> > ### Author Response · Authors · 2025-11-21
> >
> > **Q5. Reranking Strategy (Lines 74–83 and 308–323)**.
> >  - We would like to emphasize that reranking is not the main methodological contribution of PhenoKG. As stated in the reviewer's summary, reranking is optional; the core contributions are (i) patient-specific subgraph construction, (ii) the hybrid GNN + Transformer architecture, and (iii) the dual contrastive objective. Regarding reranking, we did include ablation experiments, reported in Tab. 4 (Appx. A). The magnitude of the gain depends on how complete the underlying KG is and if it has relevant information to this patient: in settings where important phenotype–gene associations are missing, reranking can provide a noticeable improvement, while in more complete regions of the graph, the effect is smaller.
> >
> >  - Methodologically Principled Approach: PhenoKG relies on a fixed KG, and newly validated phenotype–gene associations cannot be incorporated into pretrained embeddings without retraining the entire model. The reranking strategy provides a practical and theoretically justified alternative by integrating new evidence as a post-processing signal, avoiding inconsistencies in the embedding space and enabling updates without re-training (Sec. 3.4) . We will clarify these points in the revision.
> >
> > **Q6. Statistical Significance Measures** (Line 294). PhenoKG outputs similarity-based ranking scores, which reflect relative distances in the embedding space for each patient–gene pair (Sec. 3.4). These scores are not derived from a statistical test or a parametric model, so per-patient p-values are not defined in this setting. This is consistent with prior phenotype-driven methods such as Shepherd and PhenoApt, which also report rankings rather than hypothesis-testing statistics. To quantify statistical variability, we run all experiments three times with different random seeds and report means and standard deviations for all metrics (Tab. 1–2). We will clarify this distinction in the revised manuscript.

---

### Official Review · Reviewer_KDxh · 2025-11-01

**Soundness:** 3
**Presentation:** 3
**Contribution:** 1
**Rating:** 2
**Confidence:** 3

**Summary:**

This paper introduces a new knowledge graph for phenotype. From a scientific, and even from a machine learning perspective, it is useful to have more data sources like this. However, the novel part of the paper is the dataset itself. The machine learning method itself is a combination of well-known approaches, and I do not identity novel insights.

**Strengths:**

* The presented dataset is an interesting one
* The experiments appear to be performed correctly

**Weaknesses:**

* I see little contribution to machine learning research.

**Questions:**

* Are there specific machine learning contributions I might have overlooked?

**Details Of Ethics Concerns:**

There is no ethics statement in the paper, but the paper is about genes and phenotypes which to my understanding are patient data.

---

> ### Author Response · Authors · 2025-11-21
> **Rebuttal by Authors**
>
> We thank the reviewer for taking the time to review our submission. We would like to clarify a key misunderstanding in order to more accurately reflect the contribution of the work.
>
> **Q1. Dataset and Knowledge Graph.** We do not introduce a new knowledge graph or a new phenotype dataset. PhenoKG uses PrimeKG\[a\] Knowledge Graph(KG) (Sec. 4.1), and the patient-specific subgraphs are only a derived preprocessing step, not a new KG or data source.
>
> **Q2. Technical Contributions.** The contribution of the paper lies in the proposed gene prioritization framework, which introduces several methodological elements that, to our knowledge, have not been combined or evaluated in this context:
>
> * Patient-specific subgraph construction enabling inference without curated candidate lists (Sec. 3.2).
> * A hybrid GNN \+ Transformer architecture to jointly model phenotype–gene structure (Sec. 3.2.1).
> * A Contrastive and Patient Similarity Losses (Sec. 3.3; Appx. A).
> * A dynamic inference-time reranking mechanism incorporating external phenotype–gene associations to extend the existing in KG information (Sec. 3.4).
>
> These components lead to consistent state-of-the-art performance across multiple real-world datasets (Tab. 1–2).
>
> **Q3. Ethics**. Our work uses only public, fully de-identified datasets, and does not involve any human-subject data collection, annotation, intervention, or data release (Sec. 4.1) .
> An ethics statement can be added for clarity if the reviewers and AC find it beneficial. We appreciate the reviewer’s comments and hope this explanation helps clarify the scientific contribution of the work.
>
> \[a\] Chandak, P., Huang, K., & Zitnik, M. (2023). Building a knowledge graph to enable precision medicine. Scientific Data, 10(1), 67\.

---

### Author Response · Authors · 2025-11-26
**Rebuttal by Authors**

Dear Reviewers,

Thank you for your valuable comments. We have added new results on filtering for the FGDD and PhenoDis datasets, and expanded the description of dataset preprocessing, subgraph construction, and dataset overlap, as well as clarifying the methodological differences from prior work. All issues highlighted in the review were addressed in the revised PDF, and the updated MyGene2 results further support our findings. Overall, the results confirm that our method performs at the same level or substantially better than the comparative approaches. Please refer to the updated PDF for details. We would be happy to address any remaining concerns if needed.

---

### Meta-Review · Area_Chair_Cmpb · 2026-01-04

**Summary:**

I carefully read the paper, and my main concern is the lack of novelty and the limited significance of the methodological contribution, which aligns with the concerns raised by two of the reviewers. I have also carefully read all of the authors’ rebuttals and acknowledge the careful design of the proposed method. However, the approach does not go beyond an engineering combination of existing techniques. While such combinations can be valuable when they lead to nontrivial results or new insights, I do not believe this work achieves that.

Although the work appears to be practically relevant and the presentation is generally clear, I recommend rejection of the paper.

**Reviewer Concerns:**

- Two reviewers raised concerns regarding the novelty and methodological contributions of the paper. Although the authors provided detailed responses to address these issues, and I acknowledge that some of the reviews were overly simplistic, I agree that this remains the main weakness of the manuscript. In my view, these concerns are still outstanding and constitute the primary reason for my decision. While the authors made several improvements during the revision, this is a fundamental issue that cannot be adequately resolved through rebuttals or further revisions.
- Issues related to scalability, consistency of the model’s performance, and practical relevance have been sufficiently addressed in the authors’ responses.

**Reviewer Scores:**

The main concern raised by **Reviewers KDxh** and **PmbE** is the lack of methodological novelty, which is fundamentally difficult to address through discussion or revision. Therefore, I believe their scores are unlikely to change.

**Reviewer Fp52**’s concerns focus on scalability and the consistency of the model’s performance. As the authors explained in their rebuttal, these issues do not constitute major weaknesses of the method, and there is therefore some possibility that the score could be increased.

**Reviewer 9P6M** raised concerns regarding practical relevance and clarity, both of which have been addressed in the rebuttal. Accordingly, I believe this reviewer may increase the score.

---

### Decision · Program_Chairs · 2026-01-26

Reject